# A Zika virus vaccine expressing premembrane-envelope-NS1 polyprotein

Anzhong Li[1], Jingyou Yu[1,2], Mijia Lu[1], Yuanmei Ma[1], Zayed Attia[1], Chao Shan [3], Miaoge Xue[1], Xueya Liang[1], Kelsey Craig[1], Nirajkumar Makadiya[1], Jennifer J. He[1], Ryan Jennings[1], Pei-Yong Shi[3], Mark E. Peeples[4,5], Shan-Lu Liu[1,2,6,7], Prosper N. Boyaka[1,6] & Jianrong Li[1,6]

Current efforts to develop Zika virus (ZIKV) subunit vaccines have been focused on pre-membrane (prM) and envelope (E) proteins, but the role of NS1 in ZIKV-specific immune response and protection is poorly understood. Here, we develop an attenuated recombinant vesicular stomatitis virus (rVSV)-based vaccine expressing ZIKV prM-E-NS1 as a polyprotein. This vectored vaccine candidate is attenuated in mice, where a single immunization induces ZIKV-specific antibody and T cell immune responses that provide protection against ZIKV challenge. Co-expression of prM, E, and NS1 induces significantly higher levels of Th2 and Th17 cytokine responses than prM-E. In addition, NS1 alone is capable of conferring partial protection against ZIKV infection in mice even though it does not induce neutralizing anti-bodies. These results demonstrate that attenuated rVSV co-expressing prM, E, and NS1 is a promising vaccine candidate for protection against ZIKV infection and highlights an important role for NS1 in ZIKV-specific cellular immune responses.

[1] Department of Veterinary Biosciences, The Ohio State University, 1925 Coffey Road, Columbus, OH 43210, USA. [2] Center for Retrovirus Research, The Ohio State University, 1925 Coffey Road, Columbus, OH 43210, USA. [3] Department of Biochemistry & Molecular Biology, Department of Pharmacology & Toxicology, and Sealy Center for Structural Biology & Molecular Biophysics, University of Texas Medical Branch, 301 University Boulevard, Galveston, TX 77555, USA. [4] Center for Vaccines and Immunity, The Research Institute at Nationwide Children's Hospital, 700 Children's Drive, Columbus, OH 43205, USA. [5] Department of Pediatrics, College of Medicine, The Ohio State University, 370W. 9th Ave., Columbus, OH 43210, USA. [6] Infectious Diseases Institute, The Ohio State University, 1925 Coffey Road, Columbus, OH 43210, USA. [7] Department of Microbial Infection and Immunity, The Ohio State University, 1925 Coffey Road, Columbus, OH 43210, USA. Correspondence and requests for materials should be addressed to J.L. (email: li.926@osu.edu)

Zika virus (ZIKV) is a mosquito-borne flavivirus that was first identified in monkeys from the Zika forest, near Lake Victoria, Uganda in 1947[1–3]. Sporadic outbreaks of ZIKV have since been reported in Africa and Asia[4]. Historically, people infected with Zika virus have no or mild symptoms including fever, rash, muscle pain, red eyes, headache, and conjunctivitis[4,5]. However, in 2015 a ZIKV pandemic began in South America, Central America, the Caribbean, and the USA, suddenly becoming a global public health issue[5]. Importantly, ZIKV from these recent outbreaks can cause Congenital Zika Syndrome (including microcephaly), Guillain-Barré syndrome, and other severe neurological disorders[6,7]. ZIKV is primarily transmitted through the bite of an infected *Aedes* species mosquito although other transmission modes such as sexual, blood transfusion, and maternal-fetal are also possible[8–10]. Currently, there is no FDA-approved vaccine or antiviral drug for ZIKV.

ZIKV is a member of the virus family Flaviviridae, which also includes other globally prevalent human pathogens such as dengue virus (DENV), yellow fever virus (YFV), West Nile virus (WNV), and Japanese encephalitis virus (JEV). The ZIKV genome encodes a single polyprotein that is cleaved post-translationally into three structural proteins (capsid, premembrane, and envelope) and seven nonstructural proteins (NS1, NS2A, NS2B, NS3, NS4A, NS4B, and NS5)[11,12]. The E protein is a type II fusion protein which mediates cellular attachment and membrane fusion, and is the target for most neutralizing antibodies (Abs). Flavivirus prM protein typically associates with E to form heterodimers and is important for proper folding of E[13–16]. Co-expression of prM and E of several flaviviruses including ZIKV results in the secretion of virus-like particles (VLPs) termed recombinant subviral particles[17–19]. The prM protein is an integral part of both virions and subviral particles, and undergoes a cleavage event during virus maturation[20]. Therefore, prM and E proteins have been the primary targets for the rational design of subunit and recombinant flavivirus vaccines. However, the NS1 protein of several flaviviruses has been shown to confer protection against flavivirus infection in animal models in the absence of detectable neutralizing antibody[21–26]. Whether immunization with the ZIKV NS1 protein has similar protective capabilities is currently unknown.

Recently, several ZIKV vaccine candidates have been reported, including nucleic acid (DNA and mRNA), inactivated virus, subunit, VLP, vectored vaccines (including adenovirus and vaccinia virus), and live attenuated vaccines[17,19,27–34]. These vaccine candidates triggered various degrees of humoral and cellular immunity and protection in rodent and/or nonhuman primate models. Among these candidates, DNA vaccine, subunit vaccine, and inactivated vaccine have been initiated for clinical trials. Currently, all ZIKV subunit, DNA, and mRNA vaccines have been targeted on the E or prM-E antigen. Although these vaccine candidates are promising, exploration of other new and highly efficacious ZIKV vaccines is needed.

Vesicular stomatitis virus (VSV) is a prototype nonsegmented negative-sense (NNS) RNA virus that belongs to the Rhabdoviridae family. VSV is a natural pathogen of livestock such as cattle and swine, as such, there is no pre-existing immunity against VSV in the human population[35,36]. VSV is an excellent platform for vaccine development. VSV can accommodate multiple foreign genes, and thus can be developed into a multivalent vaccine[35,36]. Antigens are highly expressed in both cell culture and animals by VSV, enabling the generation of strong systemic immune responses[35,37]. In response to the sudden outbreaks of Ebola virus in Africa in 2013, a VSV-based Ebola virus vaccine was tested in human clinical trials[38–41]. In general, VSV is safe in humans although high doses of VSV can cause side effects in some people including joint and muscle pain[39–41]. Importantly, the VSV-based Ebola virus vaccine was shown to be highly efficacious in protecting against Ebola virus infection in humans[39–41]. During preparation of this manuscript, Betancourt et al., reported that maternal antibody derived from female C57BL/6 mice inoculated with VSV expressing prM-E can protect offspring from lethal ZIKV infection[42]. However, whether an immunized animal can be protected against ZIKV infection is not known.

Here, we developed a methyltransferase (MTase)-defective rVSV (mtdVSV)-based ZIKV vaccine platform. We recovered a panel of rVSV expressing ZIKV prM-E-NS1, prM-E, E, E truncation mutants, and NS1. These mtdVSV-based ZIKV vaccine candidates were highly attenuated but remained effective in triggering ZIKV-specific antibody and T cell immunity in mice, and provided complete protection against ZIKV challenge in immunocompetent BALB/c and type 1 interferon receptor-deficient A129 mice. In addition, we found that NS1 protein plays a regulatory role in ZIKV-specific T cell response, and that NS1 alone can confer partial protection from ZIKV infection. Collectively, this mtdVSV-based vaccine is a promising vaccine candidate for ZIKV.

## Results

**Recovery of recombinant VSVs expressing ZIKV antigens.** To determine the feasibility of using VSV as a vaccine vector to deliver ZIKV proteins, we constructed thirteen recombinant viruses using the wild-type VSV genome as the backbone (Supplementary Fig. 1A). These constructs allowed us to compare the immunogenicity of various combinations of ZIKV wild-type and mutant proteins, all including the E protein since it is known in other flaviviruses to be the main target for neutralizing antibody (Ab)[43]. ZIKV E protein is composed of an N-terminal ectodomain, consisting of three domains (I, II, and III), and a stem and a C-terminal transmembrane (TM) domain (Supplementary Fig. 1A)[13,43]. The major neutralizing epitopes are located in the ectodomain of the E protein.

We first constructed rVSV-E which would express the full-length E protein including the stem-TM domain (504 amino acids) (Supplementary Fig. 1A). Since the exact boundary between the ectodomain and the stem-TM domain is unclear, we constructed three E truncation mutants lacking the predicted stem-TM domain in three recombinant viruses, rVSV-E404, rVSV-E414, and rVSV-E415 which express the N-terminal 404, 414, and 415 amino acids of the E protein, respectively (Supplementary Fig. 1A). Next, rVSVs expressing E and E deletions with anchor C (signal peptide) were generated (Supplementary Fig. 1A). These recombinant viruses were named rVSV-aE, rVSV-aE404, rVSV-aE414, and rVSV-aE415. We also constructed four recombinant viruses that would co-express anchor C-prM with E, or the same E deletions, as a polyprotein (Supplementary Fig. 1A). In addition, since the NS1 protein of other flaviviruses has been shown to play a role in protection[21,25,44], we constructed rVSV-prM-E-NS1 which expresses prM, E, and NS1 as a polyprotein (Supplementary Fig. 1A).

Plaques formed by rVSV were $2.72 \pm 0.17$ mm (mean ± standard deviation) in diameter, while plaques formed by rVSV expressing ZIKV antigens ranged from $1.68 \pm 0.17$ to $2.08 \pm 0.14$ mm in diameter (Supplementary Fig. 1B), suggesting that expression of these ZIKV proteins reduced the replication and spread of rVSV. Next, we compared replication kinetics of rVSV, rVSV-E, rVSV-prM-E, and rVSV-prM-E-NS1 in cell culture, and found that rVSV-prM-E-NS1 had a significant delay in replication kinetics compared to parental rVSV ($P < 0.05$, Student's $t$-test) (Supplementary Fig. 2).

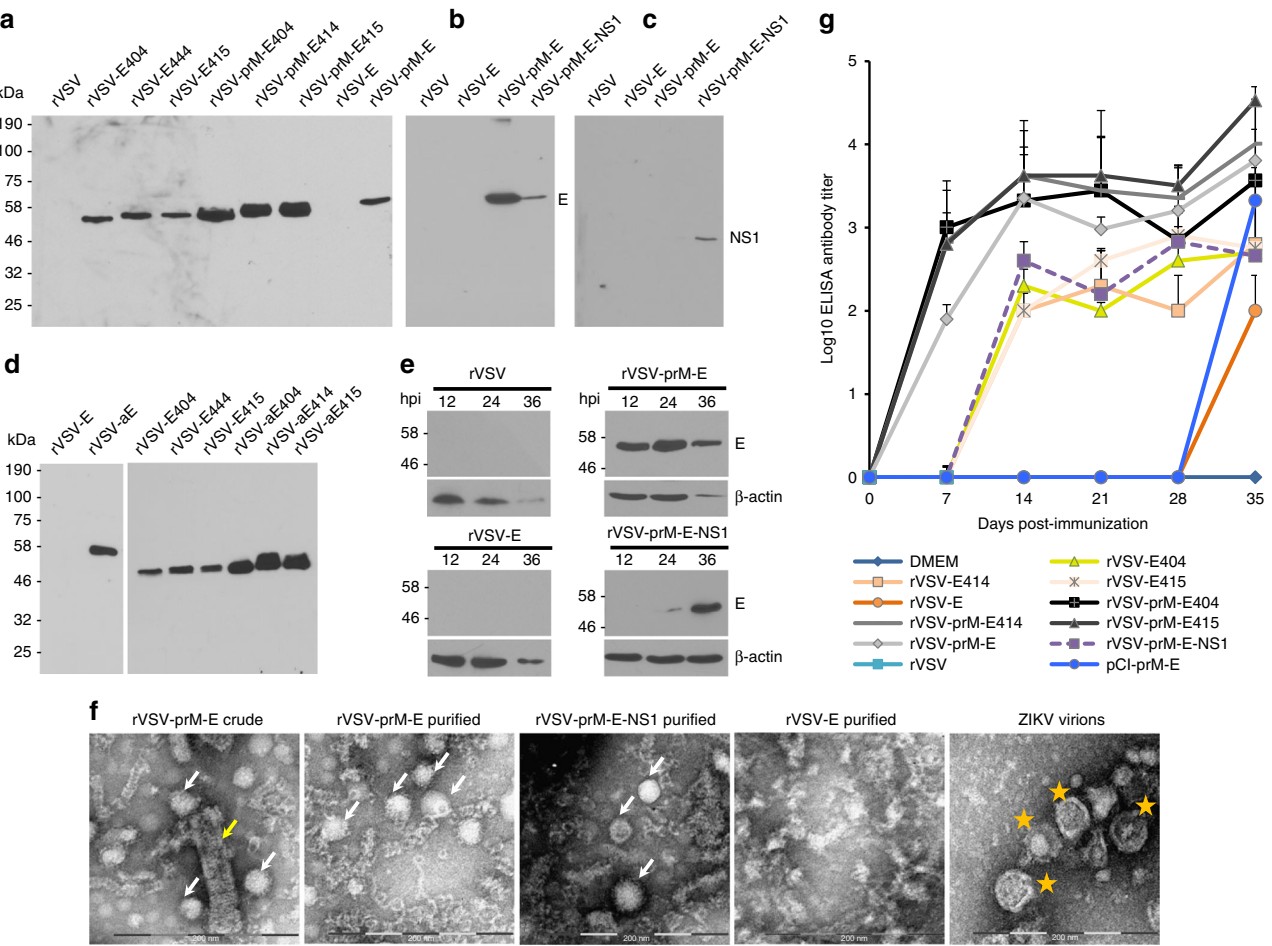

**Fig. 1** Recombinant rVSV expressing ZIKV antigens are immunogenic in mice. **a** Expression of ZIKV E truncations by VSV vector. BSRT7 cells were infected with each recombinant virus expressing ZIKV antigen at an MOI of 3.0. At 16 h post-infection, cells were lysed in 500 μl of lysis buffer, and 10 μl of lysate was analyzed by SDS-PAGE and were blotted with anti-ZIKV E protein monoclonal antibody. **b** Expression of full-length ZIKV E protein by VSV vector. BSRT7 cells were infected with the indicated recombinant virus expressing ZIKV antigen at an MOI of 3.0. Cell lysates were harvested at 16 h post-infection, and analyzed by western blot. **c** Expression of NS1 protein by VSV vector. Same cell lysates from **b** were subjected to western blot analysis using anti-ZIKV NS1 antibody. **d** Comparison of the expression of ZIKV E truncations with or without anchor C signal peptide by VSV vector. BSRT7 cells were infected with each recombinant virus at an MOI of 3.0. Cell lysates were harvested at 16 h post-infection, and analyzed by western blot. **e** Kinetics of ZIKV E protein expression by the VSV vector. Top panel: BSRT7 cells were infected with each recombinant virus at an MOI of 3.0. Cytoplasmic extracts were harvested at the indicated time points. Equal amounts of total cytoplasmic lysate were analyzed by SDS-PAGE, followed by western blot analysis. Bottom panel: Equal amounts of total cytoplasmic lysate were blotted with anti-β-actin antibody. **f** Electron microscopy analysis of ZIKV virus-like particles (VLPs). ZIKV VLPs were negatively stained with 1% ammonium molybdate and visualized by a transmission electron microscope. rVSV-prM-E crude indicates a mixture of ZIKV VLPs and VSV virions from supernatant harvested from BSRT7 cells infected by rVSV-prM-E. ZIKV VLPs were further purified from rVSV-prM-E or rVSV-prM-E-NS1-infected cells. No VLPs were found in rVSV-E. ZIKV Cambodian strain was grown in Vero cells, purified, and used as a control. The yellow arrow indicates a VSV particle; white arrows indicate ZIKV VLPs; and yellow stars indicate ZIKV virions. **g** Kinetics of ZIKV-specific ELISA antibody induced by rVSV expressing ZIKV antigens. Groups of five female BALB/c mice were inoculated intranasally with a single dose (10^6 PFU) of rVSV or rVSV expressing ZIKV antigens. For DNA vaccine, mice were immunized intramuscularly with 50 μg of pCI-prM-E, and boosted with same dose two weeks later. Serum samples were collected weekly and analyzed by ELISA for ZIKV-specific serum IgG Ab. Data are expressed as the geometric mean titers (GMT) of five mice ± standard deviation. Western blots shown are the representatives of three independent experiments

**High-level expression of ZIKV proteins by the VSV vector**. We next assessed the expression level of VSV vectored E protein and its truncations. A 54 kDa full-length E protein was detected in cells infected with rVSV-prM-E and rVSV-prM-E-NS1 but not with rVSV-E (Fig. 1a, b). The NS1 protein was only detected in rVSV-prM-E-NS1 infected cells (Fig. 1c), as expected. A smaller E protein was detected in cells expressing the truncated E protein, consistent with the shorter C-terminal domain (Fig. 1a). Quantitative analysis of three independent experiments showed that rVSVs co-expressing prM with E or E truncations had approximately five times greater E protein expression compared to rVSVs expressing E or E truncations without prM. Western blot of cell

culture supernatants showed that all rVSVs co-expressing prM and E/E truncations released enough E/E truncation proteins into the supernatant to be easily detectable without the need for concentration (Supplementary Fig. 3A, B). The expression of prM followed by truncated E likely results in the secretion of soluble E. NS1 protein was also secreted into cell culture medium (Supplementary Fig. 3C). However, no E/E truncation was detectable in cell medium from rVSVs expressing E/E truncations alone (Supplementary Fig. 3A, B). Next, we compared the expression level of E/E truncations with or without anchor C signal peptide by VSV vector. As shown in Fig. 1d, rVSV constructs with anchor C had more abundant expression of E/E truncations compared to

rVSV constructs without anchor C. E truncations were also detected in the supernatants of rVSV constructs with anchor C (Supplementary Fig. 3D). However, full-length E protein was still not detectable by rVSV-aE, even though the anchor has been fused with E (Supplementary Fig. 3D). Thus, co-expression of anchor C and prM with the E/E truncations significantly increased their E expression and/or stability. These results also indicate that the prM, E, and NS1 proteins were proteolytically cleaved from the polyprotein and secreted into cell culture supernatants.

We also determined the kinetics of E protein expression (Fig. 1e). E protein was detectable in rVSV-prM-E at 12 h post-infection, reached the highest expression level at 24 h post-infection, and declined by 36 h primarily because cells were lysed by this time point as indicated by the reduction in β-actin. Recombinant rVSV-prM-E-NS1 had a significant delay in E protein expression. E protein was detectable at 24 h and reached its highest level at 36 h post-infection, without cell death as indicated by a continued high level of β-actin. E protein was not detectable in cells infected by rVSV-E even at the time when most cells were lysed. The expression of E protein by rVSV-prM-E and rVSV-prM-E-NS1 but not by rVSV-E in virus-infected cells was confirmed by [35S] methionine-cysteine metabolic labeling (Supplementary Fig. 4).

**Expression of prM-E or prM-E-NS1 by rVSV generates VLPs.** Cells were infected with rVSV-E, rVSV-prM-E or rVSV-prM-E-NS1 and the cell culture medium was harvested at 24–48 h post-infection. Two types of particles, VSV (yellow arrow) and ZIKV VLPs (white arrow) (Fig. 1f and Supplementary Fig. 5A), were detected by negative-staining and electron microscopy. After separation by CsCl isopycnic gradient centrifugation, a large number of low density ZIKV VLPs were obtained from rVSV-prM-E and rVSV-prM-E-NS1 infected cells (Fig. 1f and Supplementary Fig. 5B, C). The ZIKV VLPs expressed by VSV had a diameter of 30–40 nm, which are relatively smaller than native ZIKV virions (40–50 nm) (Fig. 1f and Supplementary Fig. 5E, F). No VLPs were detected in cell culture medium from cells infected with rVSV-E (Fig. 1f and Supplementary Fig. 5D). Therefore, these results confirm that expression of prM-E and prM-E-NS1 but not E alone by the VSV vector resulted in the assembly of VLPs.

**Co-expression of prM with E/E truncations induce more antibody.** Next, we tested the immunogenicity of nine recombinant viruses in mice. BALB/c mice were inoculated intranasally with a single dose ($10^6$ PFU) of each recombinant virus. The DNA vaccine was used as a control. Briefly, mice were intramuscularly injected with 50 μg of pCI-prM-E and were boosted with same dose of pCI-prM-E two weeks later. Mice infected with rVSV exhibited severe clinical signs, including ataxia, hyperexcitability, and paralysis. At 7 days post-inoculation, two of the five mice were dead, and the remaining three mice were dead at day 10 post-inoculation. However, rVSVs expressing ZIKV antigens showed various degrees of attenuation. Mice inoculated with these recombinant viruses had mild clinical signs (such as a ruffled coat) and experienced body weight losses for 1 week, but started to gain weight by 10 days (Supplementary Fig. 6). Overall, rVSVs co-expressing prM and E/E truncation mutants were more attenuated in mice than rVSV expressing E/E truncation alone. For example, rVSV-prM-E and rVSV-prM-E414 had significantly less body weight loss compared to rVSV-E ($P = 0.021$, t-test) and rVSV-E414 ($P = 0.045$, t-test) respectively at day 7 post-inoculation. Recombinant rVSV-prM-E-NS1 was the most attenuated virus

(Supplementary Fig. 6). Mice inoculated with this virus experienced little or no weight loss and did not display any other clinical signs. This experiment demonstrated that rVSV expressing ZIKV antigens, particularly rVSV-prM-E-NS1, were significantly attenuated in mice compared to the parental rVSV.

The dynamics of ZIKV E-specific Ab production following vaccination, determined by ELISA, is summarized in Fig. 1g. At 1 week post-inoculation, most (3 or 4 out of 5) mice inoculated with rVSV co-expressing prM-E/E truncation mutants had high levels of serum IgG against ZIKV E protein. At week 2 post-inoculation, all mice in these groups had developed IgG Ab. Ab titers further increased and remained at a high level through week 5. In contrast, none of the mice vaccinated with rVSV expressing E/E truncation mutants without prM had detectable ZIKV-specific antibody by week 1. The same was true for rVSV-prM-E-NS1. At week 2, Ab was observed in these groups and increased through week 5. However, the Ab titers in these groups were significantly lower than those of the viruses co-expressing prM-E/E truncation mutants ($P < 0.05$, t-test). Ab was not detectable in the DNA vaccine group until week 5, despite the fact that these mice had been given two doses (at week 0 and 2). The Ab detected in the DNA vaccine group at week 5 post-immunization was also lower than that induced by the rVSV-prM-E/E truncations. These results demonstrate that a single-dose inoculation of mice with rVSV co-expressing prM-E/E truncations triggered high levels of serum antibody response as early as 1–2 weeks post-inoculation.

**Attenuation of recombinant VSV expressing ZIKV antigens.** To enhance the safety of VSV as a vector, we further attenuated it with a specific mutation that inhibits its mRNA cap methyl-transferase (MTase) activity. We previously showed that a single point mutation (G1670A) in the S-Adenosyl methionine (SAM) binding site in the MTase region of the large (L) polymerase protein resulted in a recombinant virus (rVSV-G1670A) that was defective in mRNA cap guanine-N-7 methylation but not ribose 2′-O methylation[45]. Compared to rVSV, this recombinant virus was highly attenuated in cell culture as well as in mice[45,46]. The G1670A mutation was introduced into VSV backbone to generate rVSV-G1670A-E, rVSV-G1670A-aE, rVSV-G1670A-prM-E, and rVSV-G1670A-prM-E-NS1, respectively (Supplementary Fig. 7A). The plaque diameters of rVSV-G1670A-E, rVSV-G1670A-aE, rVSV-G1670A-prM-E, and rVSV-G1670A-prM-E-NS1 were $1.64 \pm 0.08$, $1.68 \pm 0.10$, $1.73 \pm 0.11$, and $0.85 \pm 0.12$ mm respectively, significantly smaller than the recombinant viruses derived from the wild-type VSV backbone (Compare Supplementary Fig. 7B, Fig. 1b). Single-step replication curves showed that VSV-G1670A-E, VSV-G1670A-aE, and rVSV-G1670A-prM-E had replication kinetics similar to rVSV-G1670A, whereas rVSV-G1670A-prM-E-NS1 had a significant delay ($P = 5.82 \times 10^9$ and $0.0021$ at time points 12 and 24 h respectively, t-test) (Fig. 2a). At 24 h post-infection, the expression of E can be ranked rVSV-G1670A-prM-E > rVSV-G1670A-prM-E-NS1 > rVSV-G1670A-aE (Fig. 2b). No E protein expression was detected from rVSV-G1670A or rVSV-G1670A-E, but NS1 protein expression was detected from rVSV-G1670A-prM-E-NS1 (Fig. 2b). In a kinetic experiment, E protein expression was maximal from rVSV-G1670A-aE and rVSV-G1670A-prM-E at 12 and 36 h respectively, but was delayed in rVSV-G1670A-prM-E-NS1-infected cells (Fig. 2c). Similarly, E and NS1 proteins were secreted into cell culture medium in virus-infected cells (Fig. 2d, e, and Supplementary Fig. 8). Compared to the parental rVSV vector, E protein expression was delayed from the rVSV-G1670A vector (Fig. 2d, compare upper and lower panels), suggesting that

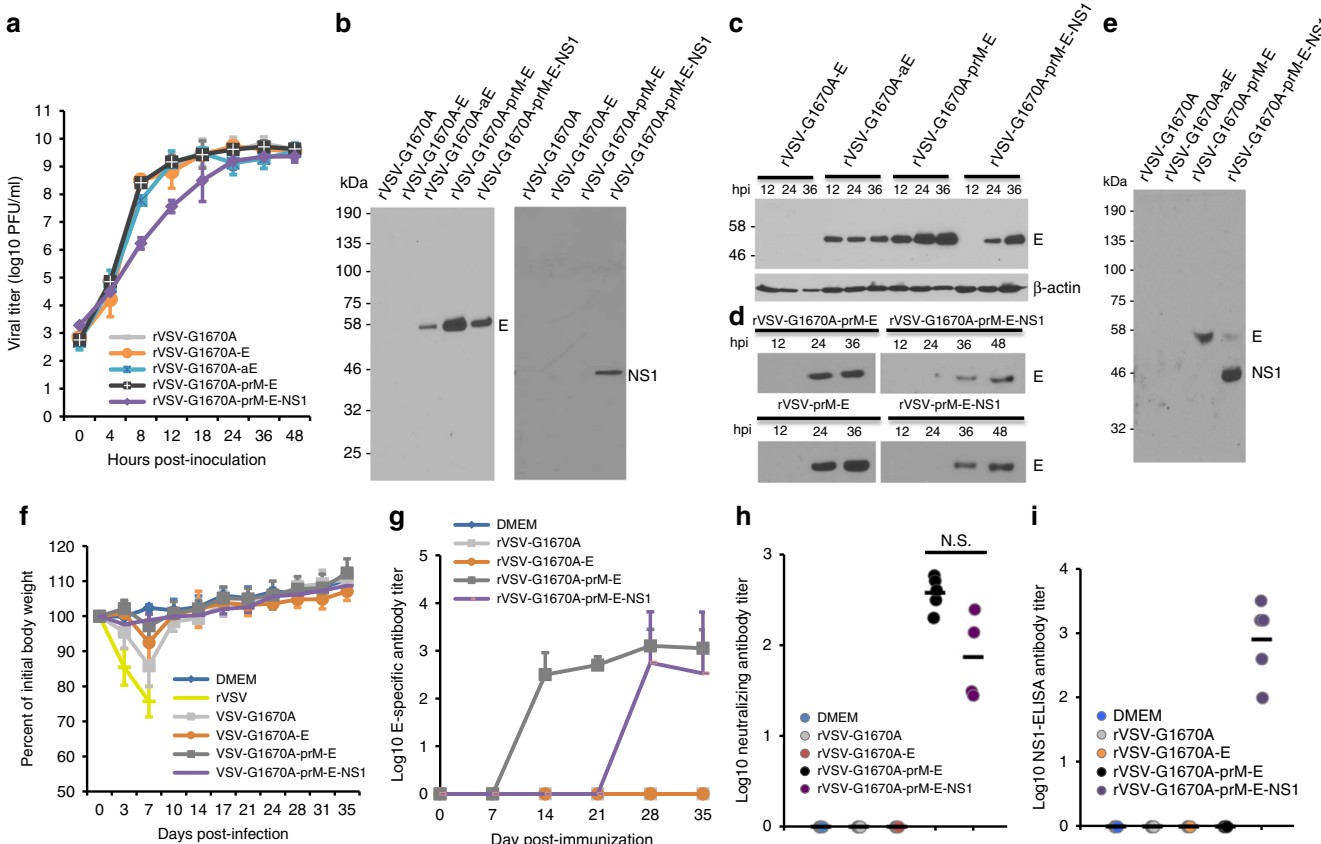

**Fig. 2** ZIKV antigen expression and antibody response by MTase-defective rVSV (mtdVSV) vector. **a** Single-step growth curve of mtdVSVs. Confluent BSRT7 cells were infected with individual viruses at an MOI of 3.0. After 1 h of incubation, the inoculum was removed, the cells were washed with DMEM, and fresh medium (containing 2% fetal bovine serum) was added, followed by incubation at 37 °C. Samples of supernatant were harvested at the indicated intervals over a 48-h time period, and the viral titer was determined by plaque assay. Data are the GMT of three independent experiments ± standard deviation. **b** Expression of ZIKV antigens by the mtdVSV in cell lysates. BSRT7 cells were infected with each recombinant virus at an MOI of 3.0. At 24 h post-infection, cell lysates were harvested and analyzed by western blot using antibody against ZIKV E or NS1 protein. **c** Kinetics of ZIKV E protein expression in cell lysates. Top panel: BSRT7 cells were infected with each recombinant virus at an MOI of 3.0. At 12, 24, and 36 h post-infection, cell lysates were harvested and analyzed by western blot using antibody against ZIKV E protein. Bottom panel: Equal amounts of total cytoplasmic lysate were blotted with anti-β-actin antibody. **d** Kinetics of ZIKV E protein release into cell culture supernatants. Cell culture supernatants were harvested from virus-infected cells at the indicated time points, and 10 μl of supernatant was analyzed by western blot using E-specific antibody. **e** ZIKV NS1 protein released into the cell culture supernatant. Cell culture supernatants were harvested from virus-infected cells at 36 h post-infection, and 10 μl of supernatant was analyzed by western blot using ZIKV serum antibody. **f** Dynamics of mouse body weight after inoculation with mtdVSV. Five six-week-old female BALB/c mice in each group were intranasally inoculated with DMEM or $10^6$ PFU of rVSV or mtdVSV expressing ZIKV antigens. The body weight for each mouse was evaluated at indicated time points. The average body weights of five mice were shown. All mice in rVSV group were dead and euthanized at day 7. **g** Kinetics of ZIKV specific antibody induced by mtdVSV expressing ZIKV antigen. Serum samples were collected weekly and analyzed by ELISA for ZIKV-specific serum IgG Ab. The titers are expressed as the GMT of five mice ± standard deviation. **h** ZIKV specific neutralizing antibody titer at week 5 post-inoculation. **i** ZIKV NS1-specific antibody detected by ELISA at week 5 post-inoculation. The western blot gels presented are a representative of three independent experiments. Mouse body weights are mean of five mice ± standard deviation

the rVSV-G1670A vector was more attenuated. These results demonstrated that ZIKV E and NS1 proteins were highly expressed by MTase-defective rVSV.

**mtdVSV-based vaccines are highly attenuated and immunogenic**. We next tested the MTase-defective VSV (mtdVSV)-based vaccines in BALB/c mice. Intranasal wild-type rVSV killed the mice within 7 days (Fig. 2f). Mice inoculated with rVSV-G1670A or rVSV-G1670A-E showed 13 and 7% weight loss at day 7 post-infection but both recovered by day 10 (Fig. 2f). Mice inoculated with rVSV-G1670A-prM-E-NS1 or rVSV-G1670A-prM-E exhibited 1–2% body weight loss but were not significantly different with DMEM control ($P > 0.05$, $t$-test) (Fig. 2f) or VSV-

associated clinical symptoms, indicating high degree of attenuation.

High levels of ZIKV E-specific antibody were detected by ELISA in rVSV-G1670A-prM-E and rVSV-G1670A-prM-E-NS1 mice at weeks 2 and 4 post-immunization, respectively (Fig. 2g). There was no significant difference in ELISA (Fig. 2g) or neutralizing antibody titer (Fig. 2h) at week 5 between these two groups ($P > 0.05$). No ZIKV specific antibody was detected in DMEM, rVSV-G1670A or rVSV-G1670A-E groups. Compared to the wild-type rVSV backbone, mtdVSV-based viruses had a delayed antibody response (compare Fig. 1f, Fig. 2g), reflecting the significant more attenuation of these recombinant viruses. In addition, all mice in the rVSV-G1670A-prM-E-NS1 group developed NS1-specific antibody as detected by ELISA at week

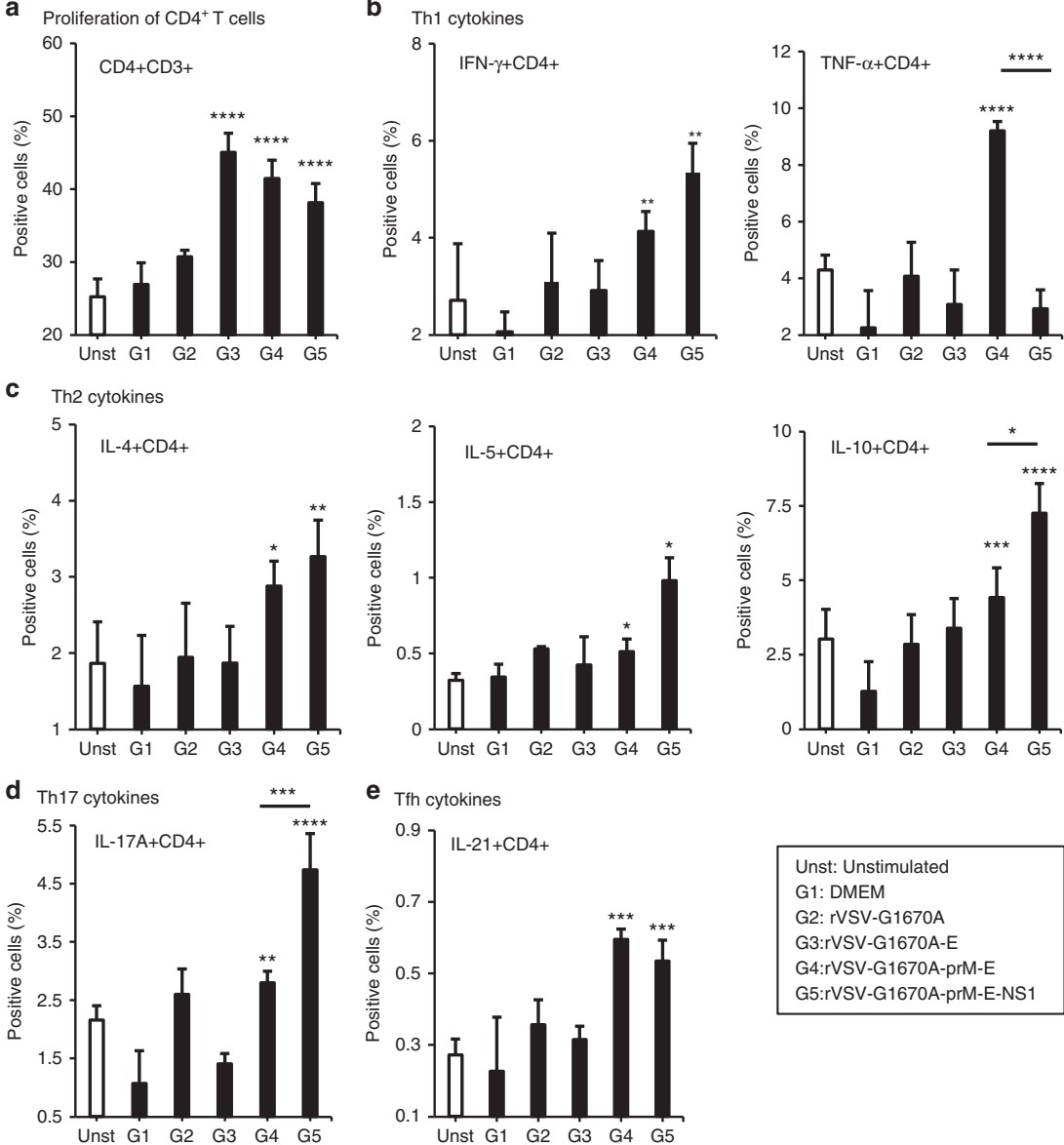

**Fig. 3** MTase-defective rVSV (mtdVSV)-based vaccine induces ZIKV-specific T helper cell responses. Six-week-old BALB/c mice were immunized with each vaccine candidate (5 mice per group). Mice were euthanized at day 35 post-immunization, the spleen was isolated from each mouse, homogenized, a cell suspension prepared, split into three wells (triplicate per mouse) and cultured in 96-well microtiter plates in the presence of 20 μg/ml of ZIKV E protein for 5 days. **a** Proliferation of CD4+ T cells. The frequencies of ZIKV-specific Th1 (IFN-γ+CD4+ and TNF-α+CD4+) (**b**), Th2 cells (IL-4+CD4+, IL-5+CD4+) (**c**), Th17 (IL-17A+ CD4+) (**d**), and Tfh (IL-21+ CD4+) (**e**) cells were determined by flow cytometry after intracellular staining with the corresponding anti-cytokine. Data were expressed as mean % positive cells (the mean of 15 samples: 3 wells × 5 mice) ± standard deviation. Asterisk indicates that the group was statistically different with unstimulated and DMEM groups. *P*-value in from left to right for each panel by Student's *t*-test: **a** ****$P = 3.55 \times 10^{-9}$, ****$P = 4.10 \times 10^{-6}$, ****$P = 4.21 \times 10^{-7}$. **b** **$P = 0.00676$, **$P = 0.00394$, ****$P = 7.58 \times 10^{-6}$, ****$P = 3.32 \times 10^{-5}$. **c** *$P = 0.0243$, **$P = 0.00180$, *$P = 0.0304$, *$P = 0.0149$, ***$P = 0.000409$, ****$P = 7.72 \times 10^{-6}$, *$P = 0.0102$. **d** **$P = 0.00749$, ****$P = 2.52 \times 10^{-6}$, ***$P = 0.000907$. **e** ***$P = 0.000313$, ***$P = 0.000162$

5 (Fig. 2i). These results demonstrated that mtdVSV-based ZIKV vaccine candidates are highly attenuated and immunogenic in mice.

Co-expression of NS1 regulates ZIKV-specific T cell responses: Induction of antigen-specific Ab and cytotoxic T cell responses capable of providing protection after immunization requires T helper cells (CD4+CD3+ cells). We found that spleen cells from mice that had been intranasally immunized with rVSV-G1670A-E, rVSV-G1670A-prM-E or rVSV-G1670A-prM-E-NS1 and restimulated in vitro with ZIKV E protein, increased the number of T helper cells (CD3+CD4+) (Fig. 3a). This finding indicates

that immunization induced ZIKV E-specific T cells capable of proliferation after re-exposure to the E antigen.

Th1 cells produce important cytokines (i.e., IFN-γ and TNF-α) for the production of complement fixing Abs and cytotoxic T cells, which together are crucial for protection against intracellular pathogens such as viruses. Flow cytometry analysis of CD3+CD4+ cells producing Th1 cytokines revealed that only cells isolated from mice immunized with rVSV-G1670A-prM-E and rVSV-G1670A-prM-E-NS1 expressed ZIKV antigen-specific IFN-γ producing T helper cells (CD4+IFN-γ+) (Fig. 3b). Interestingly, TNF-α producing T helper cells (CD4+TNF-α+)

were detected in the spleens of mice immunized with rVSV-G1670A-prM-E, but not rVSV-G1670A-prM-E-NS1. These results indicate that co-expression of NS1 enhances IFN-γ, but inhibits production of TNF-α by T helper cells.

Th2 cells produce an array of cytokines, which support the production of Abs more likely to protect against extracellular pathogens such as viruses. Interleukin 21, the signature product of follicular T helper cells (Tfh) and IL-17A the product of Th17 cells facilitate antibody production and affinity maturation. Both rVSV-G1670A-prM-E-NS1 and rVSV-G1670A-prM-E induced a similar level of CD4+IL-4+, a Th2 cytokine ($P > 0.05$, $t$-test) in spleen cells after in vitro restimulation with ZIKV E protein (Fig. 3c). However, rVSV-G1670A-prM-E-NS1 induced significantly higher CD4+IL-5+ and CD4+IL-10+, the other two Th2 cytokines ($P < 0.05$, $t$-test). Interestingly, rVSV-G1670A-prM-E-NS1 also induced a significantly higher Th17 response (CD4+IL-17A+) than rVSV-G1670A-prM-E ($P < 0.001$, $t$-test) (Fig. 3d). In addition, ZIKV E-specific Tfh cells (CD4+IL-21+) were produced at similar levels in rVSV-G1670A-prM-E-NS1 and rVSV-G1670A-prM-E inoculated mice ($P > 0.05$, $t$-test) (Fig. 3e). These results demonstrated that mtdVSV-based vaccines triggered ZIKV-specific T cell responses and that co-expression of NS1 enhances Th2 and Th17 responses. In addition, the fact that co-expression of NS1 enhances IFN-γ suggests that NS1 modulated the Th1 response (Fig. 3b). Collectively, our results suggest that the presence of NS1 leads to a more balanced response including Th1, Th2, and Th17 cells.

**mtdVSV-based vaccines protect BALB/c mice from viremia.** We next determined the protective effect of mtdVSV-based ZIKV vaccines in both female and male BALB/c mice. Mice were vaccinated intranasally with a single dose ($10^6$ PFU) of each recombinant virus, and were challenged intravenously with $1 \times 10^6$ PFU of ZIKV Cambodian strain (FSS13025) at week 5 post-immunization. DNA vaccine (pCI-prM-E) was used as a control and was given intramuscularly twice (at week 0 and 2). Similar to the previous observation shown in Fig. 2f, rVSV-G1670A-prM-E-NS1 was the most attenuated virus, experiencing no weight losses (Supplementary Fig. 9). All other recombinant viruses showed 9–15% weight loss at early time points but weights were recovered at day 14 (Supplementary Fig. 9). Previously, it was shown that the administration of anti-IFNAR1 antibody can render BALB/c mice more susceptible to infection of a mouse-adapted African ZIKV strain (Dakar 41519)[18]. Similarly, we passively transferred 1.8 mg of a blocking antibody, anti-IFNAR1, to each mouse 24 h prior to intravenous challenge with the ZIKV Cambodian strain. After ZIKV challenge, mice were monitored for 4 weeks. Unexpectedly, no significant weight loss or clinical symptoms were observed in any group including the unvaccinated but challenged controls (Supplementary Fig. 10).

The dynamics of viremia were monitored every 3–4 days until day 24 after ZIKV challenge (except the pCI-prM-E group which was only monitored at days 3 and 7) and detected by real-time RT-PCR. For the unvaccinated challenged controls, the peak of viremia was observed at day 3, declined by days 7 and 10, and cleared by day 14 (Fig. 4a). This was consistent with previous observations that ZIKV only causes transient viremia in BALB/c mice[17,30]. Similarly, mice in the rVSV-G1670A and rVSV-G1670A-E groups developed viremia, shedding an average of 3.7 logs of ZIKV PFU equivalent RNA/ml in blood samples collected at day 3 post-challenge (Fig. 4a). In contrast, viremia in most mice vaccinated with rVSV-G1670A-prM-E, rVSV-G1670A-prM-E-NS1, and pCI-prM-E groups were under the detection limit at day 3 (3 and 4 mice in rVSV-G1670A-prM-E and rVSV-G1670A-prM-E-NS1 had near detection limit level of viremia,

respectively, and 1 mice in pCI-prM-E group had a high level of viremia) (Fig. 4a and Supplementary Fig. 11). In addition, viremia was under detection limit from days 7 to 24 in these groups (Fig. 4a). Collectively, these data show that a single dose vaccination of mtdVSV-based vaccines provides protection against ZIKV-induced viremia in BALB/c mice. To determine whether VSV was persistent in the vaccinated mice, brain tissues were collected at the termination of the study for detection of VSV. No infectious VSV was detected by plaque assay in any brain tissues in any group. However, 4–5 log VSV RNAs were detected in the brains of the rVSV-G1670A, rVSV-G1670A-E, and rVSV-G1670A-prM-E groups (Fig. 4b). In contrast, nearly no VSV RNA was detected in the rVSV-G1670A-prM-E-NS1 group (Fig. 4b). Therefore, rVSV-G1670A-prM-E-NS1 is the most attenuated of these viruses.

Next, we repeated the above animal experiment and included rVSV-G1670A-aE in the vaccination. Recombinant rVSV-G1670A-prM-E-NS1 and rVSV-G1670A-aE had no body weight loss whereas rVSV-G1670A-prM-E had approximately 4.2% body weight loss at day 7 (Supplementary Fig. 12). High E-specific antibodies were observed in all animals vaccinated with rVSV-G1670A-prM-E or rVSV-G1670A-prM-E-NS1 at day 28 and further increased at day 35 post-vaccination (Fig. 4c). Only 1 out of 5 animals vaccinated with rVSV-G1670A-aE developed E-specific antibodies from day 7 to 28, and all animals developed E-specific antibodies at day 35 (Fig. 4c). NS1-specific antibodies were only detected in rVSV-G1670A-prM-E-NS1 group (Fig. 4d). Upon ZIKV challenge, mice did not exhibit body weight loss (Supplementary Fig. 13). Mice vaccinated with rVSV-G1670A-prM-E and rVSV-G1670A-prM-E-NS1 were protected from viremia at days 3 (Fig. 4e) and 7 (Fig. 4f) post-challenge whereas mice received rVSV-G1670A-aE shed high titer of ZIKV RNA in blood in a level similar to the rVSV-G1670A and saline control groups.

**mtdVSV-based vaccine provides complete protection in A129 mice.** Finally, we assessed the protective effect of mtdVSV-based ZIKV vaccines in A129 mice which lack the type-I interferon receptor (IFNAR). These mice have been shown to be highly permissive for both ZIKV[47,48] and VSV infection[49,50]. In fact, A129 mice are so susceptible to wild-type VSV infection that a dose of 50 PFU is lethal[50]. To reduce side effects, an intramuscular route was used for VSV vaccination. Since mtdVSV-based vaccines were significantly attenuated, we chose a dose of $10^5$ PFU for vaccination which was 2000 times higher than the wild-type VSV lethal dose. Briefly, A129 mice were immunized intramuscularly with rVSV-G1670A-prM-E-NS1, rVSV-G1670A-prM-E, or rVSV-prM-E-NS1, and the safety and antibody response were monitored. We found that rVSV-G1670A-prM-E-NS1 was completely attenuated in A129 mice, exhibiting no body weight losses or any abnormal reactions (Fig. 5a). However, rVSV-prM-E-NS1, which lacks the VSV attenuating mutation, was virulent in A129 mice, causing 2 deaths at day 7, and morbidity by day 10 that required termination of the others. Mice immunized with rVSV-G1670A-prM-E lost 20% of their weight but recovered and remained healthy. As shown in Fig. 5b–g, all mice in rVSV-G1670A-prM-E-NS1 and rVSV-G1670A-prM-E groups developed high levels of antibody, detected by ELISA (Fig. 5b, c) and by neutralization assay (Fig. 5d, e) as early as week 1 post-vaccination. Ab titers remained high at week 3 (Fig. 5c, e). In addition, high levels of NS1-specific Ab were detected at weeks 1 and 3 in the rVSV-G1670A-prM-E-NS1 group (Fig. 5f, g). At week 4 post-immunization, each group was intraperitoneally challenged with $10^5$ PFU of the ZIKV Cambodian strain. Mice in the control, unvaccinated challenged group (immunized with the

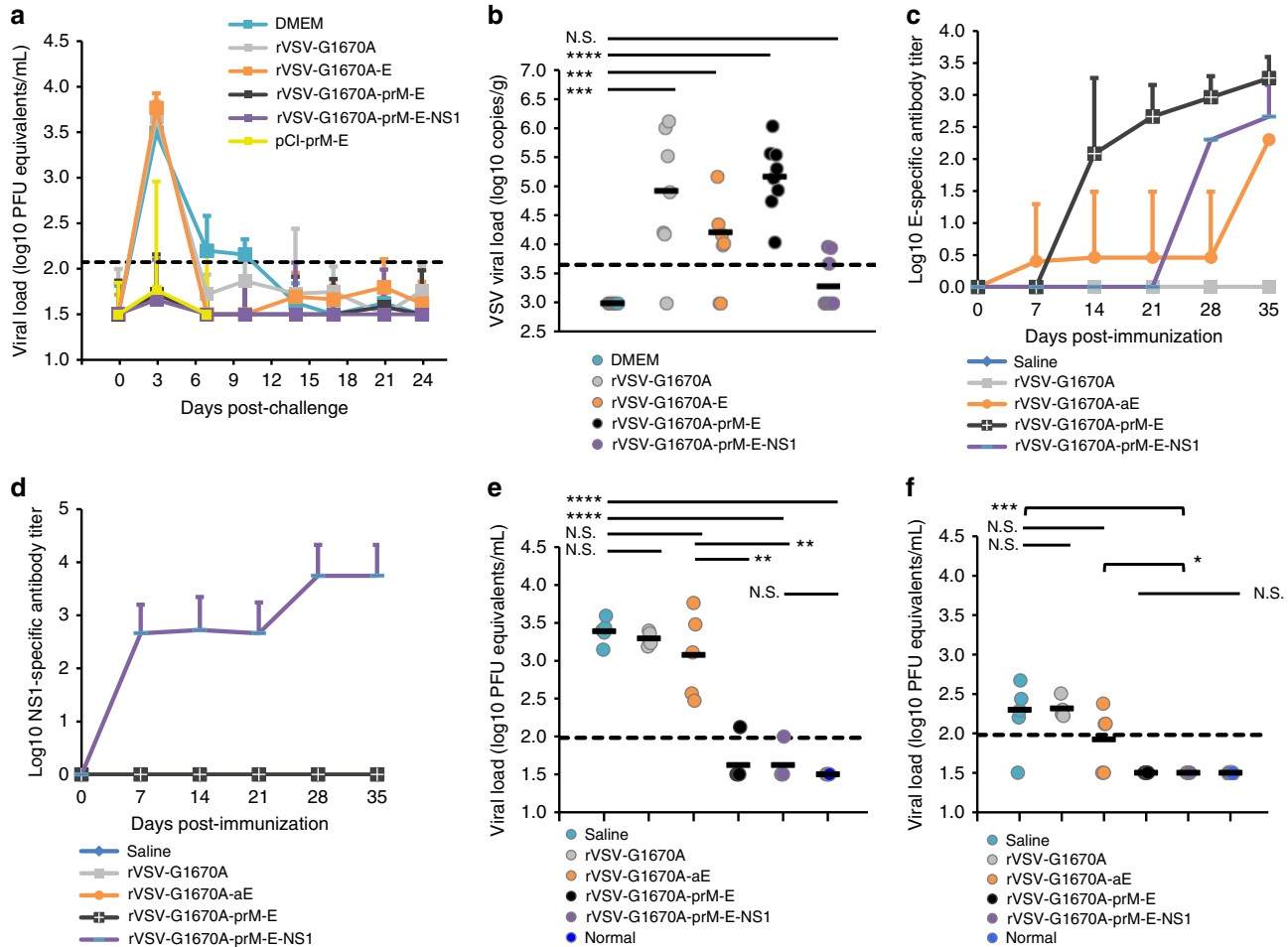

**Fig. 4** MTase-defective rVSV (mtdVSV)-based vaccine protects BALB/c mice from viremia. Mice were intranasally inoculated with DMEM or a single dose ($10^6$ PFU) of each recombinant virus. For DNA vaccine, mice were vaccinated intramuscularly with 50 μg of pCI-prM-E, and were boosted with the same dose two weeks later. At week 5, mice were intraperitoneally administered 1.8 mg of anti-IFNAR1 blocking antibody and 24 h later challenged intravenously with $10^6$ PFU of ZIKV Cambodian strain. **a** Dynamic of viremia in unimmunized mice after challenge with ZIKV. After challenge, blood samples were collected at the indicated time until day 24 and the presence of ZIKV RNA was quantitated by real-time RT-PCR and calculated to PFU equivalent RNA/ml. Data were expressed as GMT of 10 mice ± standard deviation. **b** Quantification of VSV RNA in BALB/c mice. Brains were harvested at the termination of the study. The VSV RNA was measured by real-time RT-PCR using primers annealing to the VSV L gene. Data were expressed together with the GMT of 10 mice (black bars). $P$ value (by Student's $t$-test) from top to bottom: ****$P = 4.25 \times 10^{-7}$, ***$P = 0.000710$, ***$P = 0.000371$. **c** Kinetics of ZIKV E-specific antibody induced by mtdVSV expressing ZIKV antigen. Serum samples were collected weekly and analyzed by ELISA for ZIKV-specific serum IgG Ab. Data are expressed as the GMT of five mice ± standard deviation. **d** Kinetics of ZIKV NS1-specific antibody induced by mtdVSV expressing ZIKV antigen. **e** mtdVSV-based vaccine protects BALB/c mice from viremia at day 3 post-challenge. The level of viremia was measured by real-time RT-PCR at day 3 post-challenge. $P$ value (by Student's $t$-test) from top to bottom: ****$P = 1.02 \times 10^{-5}$, ****$P = 6.06 \times 10^{-5}$, **$P = 0.00345$, ***$P = 0.00310$. **f** mtdVSV-based vaccine protects BALB/c mice from viremia at day 7 post-challenge. The level of viremia was measured by real-time RT-PCR at day 7 post-challenge. $P$ value (by Student's $t$-test) from top to bottom: ***$P = 3.89 \times 10^{-5}$, *$P = 0.0201$. Significance was calculated using $t$-test. N.S. indicates not significant

empty pCI plasmid) developed severe clinical signs (Fig. 6a) and had severe body weight loss (Fig. 6b). Because of the severity of disease in the pCI control group, these mice were terminated at day 7. In contrast, mice vaccinated with either rVSV-G1670A-prM-E-NS1 or rVSV-G1670A-prM-E did not exhibit any weight loss (Fig. 6b) or ZIKV associated clinical symptoms (Fig. 6a). ZIKV viremia was measured at days 3 and 7 post-challenge by real-time RT-PCR (Fig. 7a, b). An average of 5.8 log PFU equivalents of ZIKV was detected in the pCI control group at day 3. Low ZIKV PFU equivalents were detected at day 3 in the rVSV-G1670A-prM-E group but none in the rVSV-G1670A-prM-E-NS1 group. At day 7, high levels of ZIKV were detected in the blood of the pCI control group, whereas no or very low ZIKV was found in rVSV-G1670A-prM-E-NS1 and rVSV-G1670A-

prM-E groups. Similarly, high levels of ZIKV were detected in the brain, uterus, lung, and spleen of the pCI control group whereas under or near detection limit level of ZIKV RNA was found in these organs in the rVSV-G1670A-prM-E and rVSV-G1670A-prM-E-NS1 groups (Fig. 7c–f). In addition, histologic analysis of brain tissues showed that rVSV-G1670A-prM-E and rVSV-G1670A-prM-E-NS1 had completely protected the mice from ZIKV-induced encephalitis (Fig. 7g). In contrast, severe encephalitis characterized by neuronal necrosis, gliosis, neuronal satellitosis and neuronophagia with lymphocytic perivascular cuffing was found in the control group (Fig. 7g). The brain tissues were used for the detection of VSV RNA. It was found that VSV RNA was not detectable or near the detection limit in A129 mice vaccinated with rVSV-G1670A-prM-E-NS1 whereas

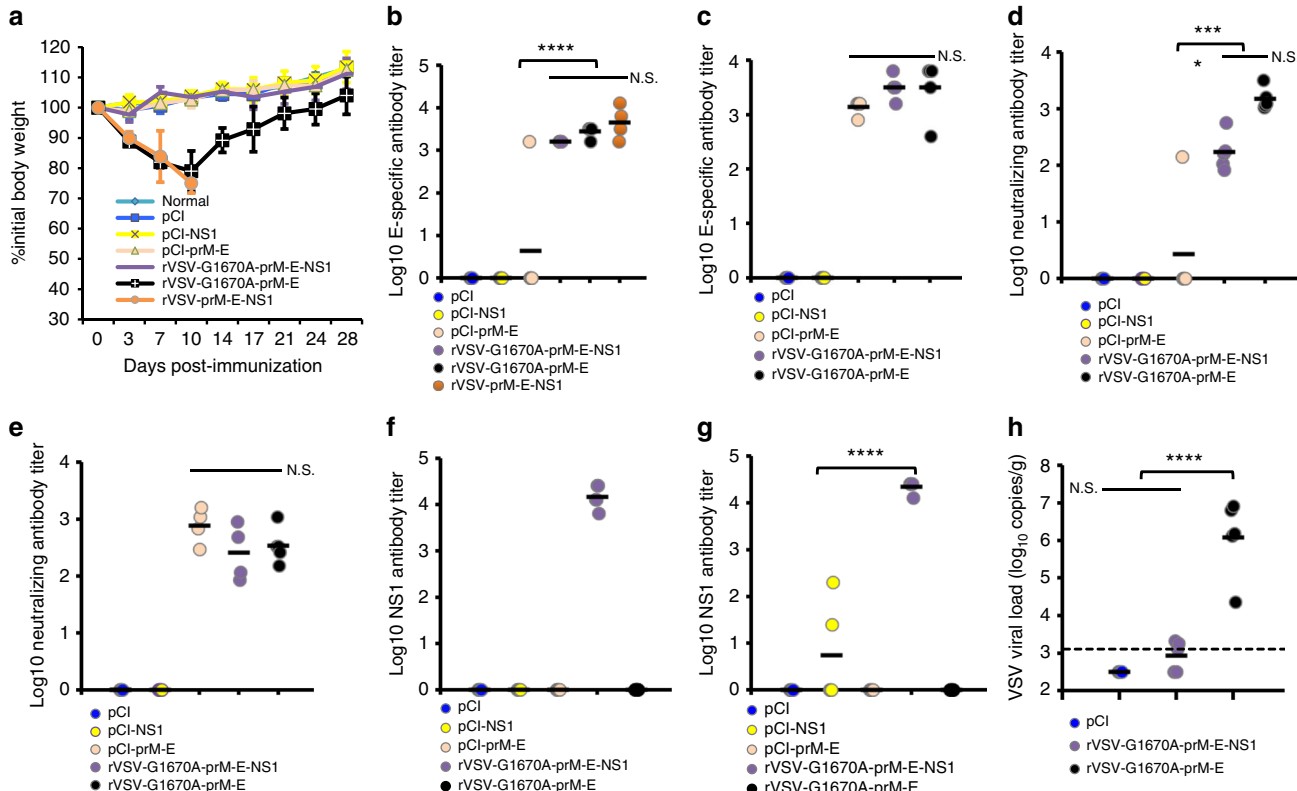

**Fig. 5** MTase-defective rVSV (mtdVSV)-based vaccine induces high levels of ZIKV-specific antibody in A129 mice. A129 mice were immunized intramuscularly with pCI, pCI-prM-E, or pCI-NS1 at a dose of 50 μg DNA per mouse, and were boosted with the same plasmid at the same dose two weeks later. For VSV-based vaccines, mice were administered intramuscularly using a single dose ($1 \times 10^5$ PFU). The body weight for each mouse was evaluated at indicated time points (**a**). The average body weights of five mice were shown. At day 7, two out of five mice in rVSV-prM-E-NS1 group were dead and the other three terminated at day 10. After immunization, blood samples were collected at weeks 1 and 3. ZIKV E-specific antibody was measured by ELISA at weeks 1 (**b**) and 3 (**c**) post-immunization. ZIKV-specific neutralizing Ab was measured at weeks 1 (**d**) and 3 (**e**) post-immunization. ZIKV NS1-specific Ab was measured by ELISA at weeks 1 (**f**) and 3 (**g**) post-immunization. ELISA titers shown are GMT of 5 mice ± standard deviation. At the termination of this experiment, brains were collected and the presence of the VSV RNA was quantified by real-time RT-PCR using primers annealing to the VSV L gene (**h**). Antibody and viral load data are expressed as the GMT of five mice (black bars). Exact P value (by Student's t-test) in each panel: **b** ****$P = 1.36 \times 10^{-6}$; **d** ****$P = 5.44 \times 10^{-5}$; **g** ****$P = 6.70 \times 10^{-5}$; **h** ****$P = 4.32 \times 10^{-6}$, N.S., not significant

approximately 6 log of VSV RNA were detected in rVSV-G1670A-prM-E group (Fig. 5h), suggesting that rVSV-G1670A-prM-E-NS1 was significantly more attenuated than rVSV-G1670A-prM-E. These data demonstrate that a single low dose of mtdVSV-based vaccines provides complete protection against ZIKV challenge in A129 mice.

**NS1 alone provides partial protection against ZIKV challenge.** We next determined whether NS1 protein alone can induce protection against a ZIKV challenge. We first used DNA vaccination approach, as we know that DNA vaccine is safe to A129 mice. The NS1 gene with anchor C signal peptide was cloned into pCI vector. Both pCI-prM-E and pCI-NS1 expressed their intended proteins, E and NS1, in transfected 293T cells (Supplementary Fig. 14). A129 mice were vaccinated intramuscularly with pCI-prM-E or pCI-NS1, and boosted with the same plasmid two weeks later. Only 1 out of 5 mice in the pCI-prM-E group had E-specific ELISA and neutralizing Ab at week 1 (Fig. 5b, d) but all of them had high levels of ZIKV E-specific Ab at week 3 (Fig. 5c, e). No ZIKV neutralizing Ab was detected in pCI-NS1 group even after the boost (Fig. 5d, e), but Ab to NS1 was detected in 2 out of 5 mice at week 3 (Fig. 5g).

Mice vaccinated with pCI-prM-E were protected from a ZIKV challenge at week 4 (Fig. 6a, b). Interestingly, 1 of the 5 mice in the pCI-NS1 group only had 10% weight loss and quickly recovered

(Fig. 6c). The other 4 mice in the pCI-NS1 group exhibited clinical signs but less severe than the pCI group although they did not have statistical difference (P > 0.05, t-test) (Fig. 6a). Overall, weight loss in the pCI-NS1 group was less than the pCI control group (Fig. 6b). At day 3 post-challenge, the pCI-NS1 group had a level of viremia similar to the pCI control group (P > 0.05, t-test) (Fig. 7a), but by day 7 the pCI-NS1 group had significantly less viremia (P < 0.05, t-test) (Fig. 7b). Similarly, significantly less ZIKV was detected in spleen, uterus, lung, and brain in the pCI-NS1 group compared to the pCI control group (P < 0.05 or P < 0.01, t-test) (Fig. 7c–f). Histologic analysis showed that the pCI-NS1 group had less severe encephalitis compared to the pCI group (Fig. 7g). Collectively, these data demonstrate that ZIKV NS1 was capable of conferring partial protection against ZIKV challenge in A129 mice in the absence of detectable ZIKV neutralizing Ab.

To further improve the protection efficacy of NS1, we recovered mtdVSV expressing NS1 alone (rVSV-G1670A-NS1) in which the ZIKV NS1 gene with anchor C was inserted at the gene junction between G and L genes (Supplementary Fig. 15). Western blot showed that the NS1 expression in rVSV-G1670A-NS1-infected cells was significantly higher than pCI-NS1-transfected cells (Fig. 6d). A pilot experiment showed that rVSV-G1670A-NS1 still caused considerable weight losses in A129 mice. Thus, we decided to test the protection efficacy of rVSV-G1670A-NS1 in BALB/c mice.

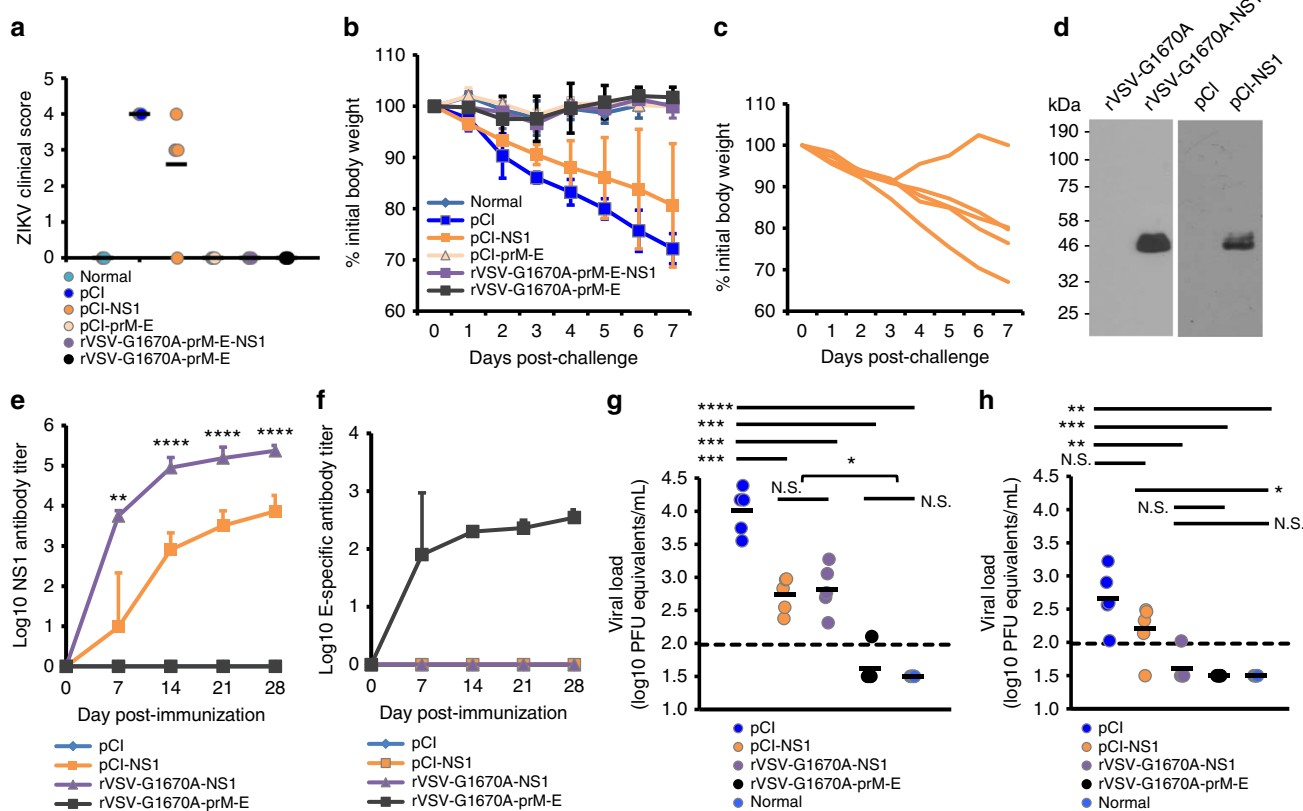

**Fig. 6** NS1 alone provides partial protection against ZIKV challenge. **a–c** Experiment in A129 mice. A129 mice were immunized with a single dose ($10^5$ PFU) of mtdVSV-based vaccine or two doses (50 μg each, two weeks apart) of DNA vaccine. At week 4 post-immunization, mice were intraperitoneally challenged with $10^5$ PFU of ZIKV Cambodian strain. **a** Clinical score in A129 mice after ZIKV challenge. ZIKV-associated clinical signs were scored for each mouse. 1 = healthy; 2 = mild: ruffled fur but no neurological symptoms; 3 = moderate: ruffled fur, hindlimb weakness, and partial hindlimb paralysis; and 4 = severe: paralysis, moribund and early removal is required. **b** Body weight change in A129 mice after ZIKV challenge. After ZIKV challenge, body weight for each mouse was evaluated daily. The average body weights of five mice ± standard deviation were shown. **c** Body weight change of A129 mice in pCI-NS1 group after ZIKV challenge. Each line represents individual animal. One mouse had 10% of body weight loss and was recovered after ZIKV challenge. **d** Expression of ZIKV NS1 protein by rVSV-G1670A and pCI vectors. BSRT7 cells were infected by rVSV-G1670A-NS1 at an MOI of 3.0 or transfected with 4 μg of pCI-NS1, and cell lysates were harvested at day 3, and subjected to western blotting using ZIKV NS1 antibody. **e–h** Experiment in BALB/c mice. 4-week-old BLAB/c mice were immunized with two doses (50 μg each, two weeks apart) of pCI-NS1, or a single dose ($10^6$ PFU) of mtdVSV-NS1 or mtdVSV-prM-E. At week 4 post-immunization, mice were administered 1.8 mg of anti-IFNAR1 blocking antibody and challenged intravenously with $10^6$ PFU of ZIKV Cambodian strain. **e** NS1-specific antibody response in BALB/c mice. $P$ value (by Student's $t$-test) from top to bottom: ****$P = 3.95 \times 10^{-5}$, ****$P = 3.48 \times 10^{-5}$, ****$P = 1.51 \times 10^{-5}$, **$P = 0.00183$. Data shown are the GMT of 5 mice ± standard deviation. **f** E-specific antibody responses in BALB/c mice. Data shown are the GMT of 5 mice ± standard deviation. **g** NS1 alone provided partial protection against viremia in BALB/c mice at day 3 post-challenge. The level of viremia was measured by real-time RT-PCR at day 3 post-challenge. $P$ value (by Student's $t$-test) from top to bottom: ****$P = 4.15 \times 10^{-6}$, ***$P = 0.000161$, ***$P = 0.000767$, ***$P = 0.000187$, *$P = 0.0250$. **h** NS1 alone provided protection against viremia in BALB/c mice at day 7 post-challenge. $P$ value (by Student's $t$-test) from top to bottom: **$P = 0.00510$, ***$P = 0.000114$, **$P = 0.00277$, *$P = 0.0308$. Data shown are the mean of 5 mice. N.S., not significant

Briefly, BALB/c mice were immunized intramuscularly with two doses (50 μg each) of pCI-NS1 or intranasally with one dose ($10^6$ PFU) of rVSV-G1670A-NS1 or rVSV-G1670A-prM-E, and were challenged intravenously with $10^6$ PFU of ZIKV at week 4 post-immunization. We found that rVSV-G1670A-NS1 triggered significantly higher NS1-specific antibody than pCI-NS1 in mice (Fig. 6e). As a positive control, recombinant rVSV-G1670A-prM-E triggered a high level of E-specific antibody (Fig. 6f). At days 3 post-challenge, mice in rVSV-G1670A-NS1 and pCI-NS1 groups had a similar level of viremia ($P > 0.05$, $t$-test) but were significantly lower than pCI control ($P < 0.05$, $t$-test) (Fig. 6g). As a positive control, the viremia level in the rVSV-G1670A-prM-E group was below or near detection limit (Fig. 6g). At day 7 post-challenge, mice in the rVSV-G1670A-NS1 and rVSV-G1670A-prM-E groups had no detectable viremia (except one in rVSV-G1670A-NS1 group which was near the detection limit) whereas mice in pCI and pCI-NS1 groups still

had a significant level of viremia ($P < 0.001$, $t$-test) (Fig. 6h). These results showed that NS1 alone was capable of triggering significant protection against ZIKV-induced viremia and that rVSV-G1670A-NS1 had a higher protection efficacy than pCI-NS1.

**Validation of the safety and efficacy of rVSV-G1670A-prM-E-NS1.** Finally, we further validated the protection efficacy of rVSV-G1670A-prM-E-NS1 in A129 mice by monitoring body weight and viremia for a prolonged time (until day 21 after challenge with ZIKV). As shown in Fig. 8a, there were no significant differences in body weight gain among three groups ($P > 0.05$, $t$-test), $10^5$ PFU of rVSV-G1670A-prM-E-NS1, saline, and normal controls, demonstrating the high safety profile of rVSV-G1670A-prM-E-NS1 in A129 mice. As expected, rVSV-G1670A-prM-E-NS1 triggered a high level of E-specific (Fig. 8b) and NS1-specific

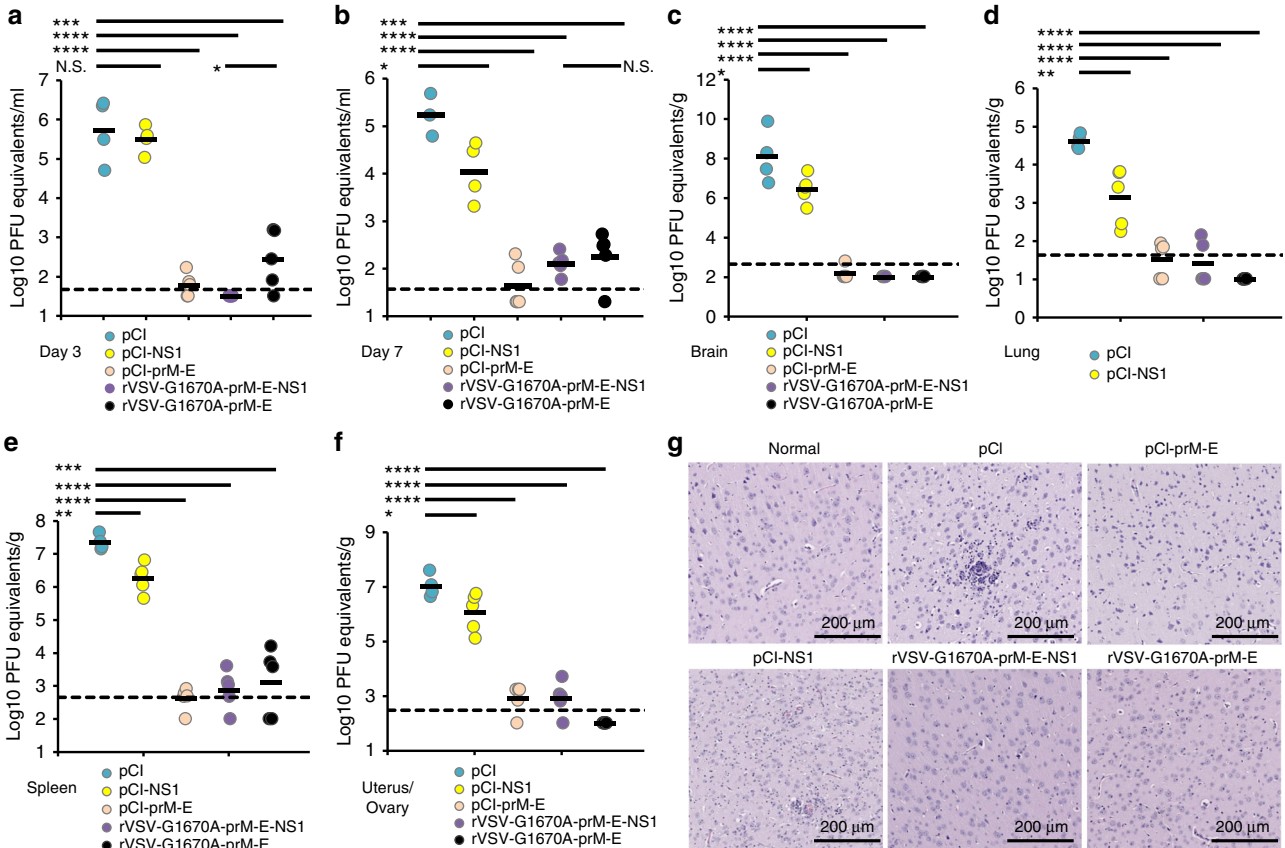

**Fig. 7** MTase-defective rVSV (mtdVSV)-based vaccine protects A129 mice from viremia and prevents ZIKV replication in vivo. After ZIKV challenge, the level of viremia in A129 mice was measured by real-time RT-PCR at days 3 (**a**) and 7 (**b**) post-challenge. At day 7 post-challenge, all mice were terminated, brain (**c**), lung (**d**), uterus (**e**), and spleen (**f**) tissues were harvested and analyzed for ZIKV RNA by real-time RT-PCR. Data shown are GMT of 5 mice. Data were analyzed using $t$-test and compared to the placebo DMEM group or the pCI group (*$P < 0.05$; **$P < 0.01$; ***$P < 0.001$; ****$P < 0.0001$; N.S., not significant, Student's $t$-test). **g** mtdVSV-based vaccine prevents ZIKV-induced encephalitis in A129 mice. Half of brain tissue from each mouse was fixed in 4% paraformaldehyde, embedded in paraffin, sectioned at 5 μm, and stained with hematoxylin-eosin (HE) for the examination of histological changes by light microscopy. Micrographs with 10× magnification (scale bar of 200 μm) are shown. $P$ value (by Student's $t$-test) from top to bottom in each panel: **a** ***$P = 0.000392$, ****$P = 6.62 \times 10^{-6}$, ****$P = 1.80 \times 10^{-5}$, *$P = 0.0236$; **b** ***$P = 0.000236$, ****$P = 1.07 \times 10^{-5}$, ****$P = 4.67 \times 10^{-5}$, *$P = 0.0383$; **c** ****$P = 2.96 \times 10^{-6}$, ****$P = 1.98 \times 10^{-5}$, ****$P = 3.46 \times 10^{-5}$, *$P = 0.0476$; **d** ****$P = 9.80 \times 10^{-9}$, ****$P = 9.95 \times 10^{-6}$, ****$P = 2.07 \times 10^{-6}$, **$P = 0.00694$; **e** ***$P = 0.000905$, ****$P = 2.31 \times 10^{-6}$, ****$P = 1.95 \times 10^{-9}$, **$P = 0.00288$; **f** ****$P = 2.85 \times 10^{-6}$, ****$P = 4.46 \times 10^{-6}$, ****$P = 9.27 \times 10^{-7}$, *$P = 0.0478$, N.S., not significant

(Fig. 8c) antibodies. Upon challenge with ZIKV Cambodian strain, mouse body weight and viremia were monitored every 1 or 3 days until day 21. Mice that received the saline control were all dead at day 6 post-challenge (Fig. 8d). The body weight in rVSV-G1670A-prM-E-NS1 group was indistinguishable with normal control ($P > 0.05$, $t$-test) at all time points (Fig. 8d). Saline control group developed high levels of ZIKV induced viremia whereas rVSV-G1670A-prM-E-NS1 group had a baseline level of viremia at day 3 and no detectable viremia between days 3 and 21 (Fig. 8e). Collectively, rVSV-G1670A-prM-E-NS1 is of high safety and efficacy against ZIKV infection.

## Discussion

The recent pandemic of ZIKV infection has already caused significant economic, health, and emotional burdens, highlighting the urgent need for developing a safe and highly efficacious ZIKV vaccine. Here, we developed an mtdVSV-based vaccine platform for ZIKV. We showed that a single dose of live attenuated VSV-based ZIKV vaccine expressing prM-E or prM-E-NS1 triggered strong ZIKV-specific neutralizing antibody and T cell immune responses, and protected BALB/c and A129 mice against ZIKV

challenge. In addition, we showed for the first time that NS1 protein modulates ZIKV-specific T cell immune responses and that NS1 alone conferred partial protection against ZIKV-induced viremia in BALB/c mice and lethal ZIKV challenge in A129 mice in the absence of a neutralizing antibody response. Altogether, these results demonstrated that this mtdVSV-based ZIKV vaccine is a highly promising candidate for ZIKV and highlighted that NS1 plays an important role in regulating immune responses against ZIKV infection.

VSV is an excellent vaccine vector[35]. It has an extremely high level of gene expression as well as rapid and efficient replication[35,51]. VSV-based vaccine is most feasible initially for controlling outbreaks of highly pathogenic agents in situations where the high risk of the pathogen would outweigh any potential side effects of the VSV vector. To date, VSV-based vaccine candidates have been developed for filoviruses (Ebola virus and Marburg virus)[38–41,52], arenaviruses (Lassa virus and LCMV)[53,54], henipaviruses (Nipah virus and Hendra virus)[55,56], and human coronaviruses (SARS and MERS)[57,58]. A single dose of these rVSV-based vaccines has been shown to highly effective in small animal models as well as nonhuman primates when

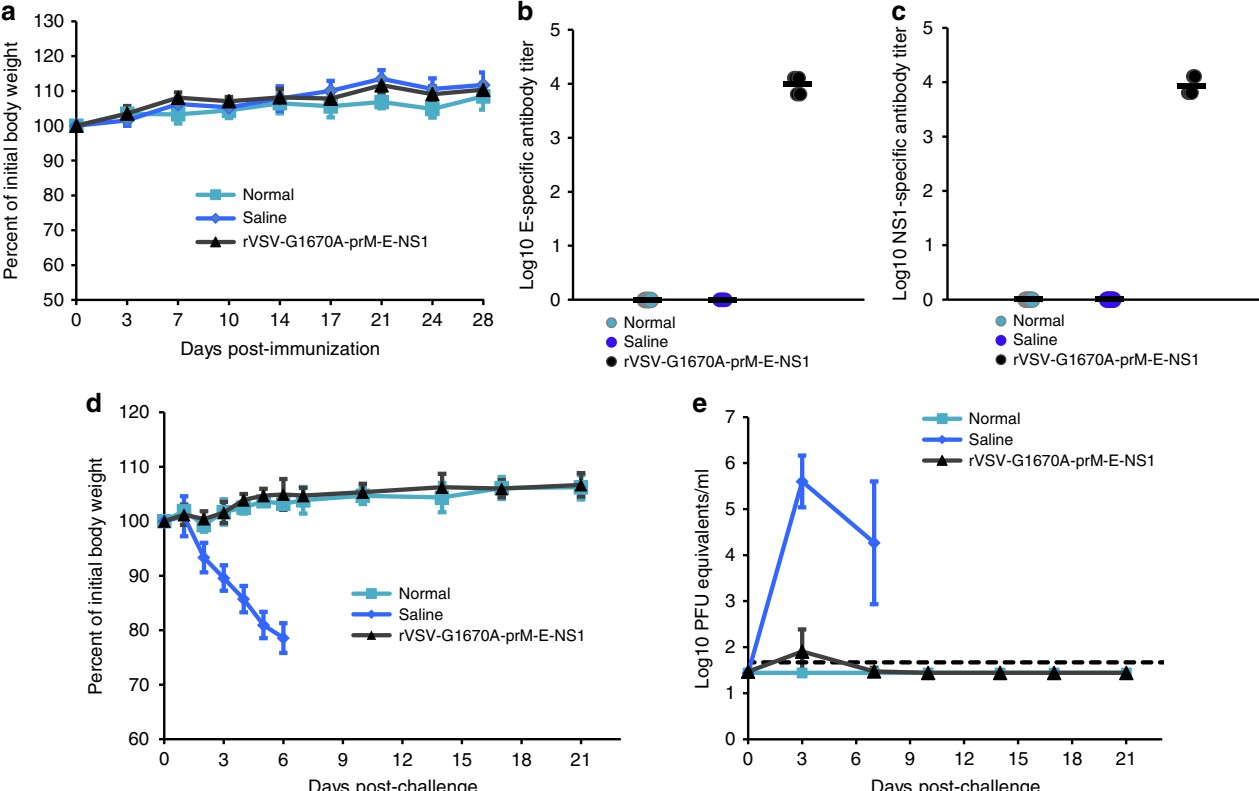

**Fig. 8** Validation the safety and immunogenicity of rVSV-prM-E-NS1 in A129 mice. A129 mice were immunized intramuscularly with $1 \times 10^5$ PFU of rVSV-prM-E-NS1 or injected with 100 µl of saline. The body weight for each mouse was evaluated at the indicated time points (**a**). Data shown are mean of 5 mice ± standard deviation. After immunization, blood samples were collected at week 4. ZIKV E-specific antibody (**b**) and NS1-specific antibody (**c**) was measured by ELISA. ELISA titers are shown as GMT of 5 mice (black bars). At week 4 post-immunization, mice were challenged with ZIKV, and their body weight (**d**) and viremia (**e**) were monitored every 1 or 3 days until day 21. Data shown are mean (**d**) or GMT (**e**) of 5 mice ± standard deviation

administered by the IN, IM, intradermal, or oral routes[35]. We now added the rVSV-based ZIKV vaccine candidate to this list. Although it was recently reported that rVSV with mutated matrix protein expressing prM-E was capable of generating maternal antibody that can protect offspring from ZIKV infection[42], it is not known whether immunized animals can be directly protected from ZIKV challenge. In our study, we systematically screened the immunogenicity of rVSVs expressing a variety of ZIKV antigens including E, E truncations, aE, aE truncations, prM-E, prM-E truncations, prM-E-NS1, and NS1. To our knowledge, this is first time to utilize co-expression of NS1 with prM-E as potential antigens in vectored ZIKV vaccine candidates. In addition, we used viral MTase as a novel approach to attenuate VSV for delivering ZIKV vaccine. We demonstrated that a single immunization (IN or IM) with mtdVSV-based vaccine triggered high levels of ZIKV-specific Ab and T cell immune responses and provided complete protection against ZIKV challenges in both BALB/c and highly sensitive A129 mice. In contrast, DNA vaccine requires high doses and multiple booster vaccination, and antibody was undetectable in both BALB/c and A129 mice at early time points (Figs. 1, 5, and Supplementary Table 1). Our results suggest that the efficacy of this VSV-based vaccine is much higher than a DNA vaccine. A VSV-based vaccine offers many other advantages, including genetic stability, expression of multiple antigens, low cost, simplicity of production, multiple routes of administration, and ease of manipulation. During the Ebola virus outbreaks in Africa in 2013–2016, VSV-based vaccine (rVSV-EBOV) was successfully used in humans and was found to be safe and efficacious[38–41]. In addition, several other VSV-based vaccine candidates and oncolytic agents are being tested in

human clinical trials. It is likely that more analysis of the safety and efficacy of VSV as a vector in humans will be forthcoming soon, which will facilitate the use of VSV-based vaccines in mass vaccination in the future.

We found that ZIKV NS1 alone can induce partial protection against ZIKV challenge. In BALB/c mice, both rVSV-G1670A-NS1 and pCI-NS1 provided substantial protection against viremia in the absence of neutralizing antibody. Overall, the protective efficacy of mtdVSV-based NS1 vaccine (rVSV-G1670A-NS1) was higher than the DNA vaccine (pCI-NS1) approach. The partial protection of pCI-NS1 against lethal ZIKV challenge was also observed in A129 mice. Overall, the disease severity, viremia, and virus burden in pCI-NS1 vaccinated group were significantly lower than the pCI control group in A129 mice ($P < 0.05$), $t$-test (Figs. 6, 7). The flavivirus NS1 is a nonstructural glycoprotein that is essential for viral replication and immune evasion. Previously, it was shown that immunization with purified NS1 and recombinant NS1 or passive administration of monoclonal antibodies (MAbs) against NS1 can protect mice against lethal infection by WNV, JEV, DENV, and YFV[23,25,44,59,60]. In addition, vaccination with DENV1, DENV3, or DENV4 NS1 provided substantial cross protection against a heterologous DENV2 lethal challenge[3]. These studies highlight the value of including the NS1 protein in candidate vaccines for flaviviruses. During revision of our manuscript, Brault et al. reported that intramuscular immunization of immunocompetent mice with the Modified Vaccinia Ankara (MVA) vector expressing NS1 (MVA-ZIKV-NS1) vaccine candidate provided 100% protection against a lethal intracerebral dose of ZIKV (strain MR766)[61]. However, their

study did not evaluate the efficacy of MVA-ZIKV-NS1 in A129 mice which is more highly sensitive to ZIKV infection.

The finding that NS1 conferred partial protection against ZIKV infection highlights an alternative strategy to design ZIKV vaccine. Current efforts on development of ZIKV subunit vaccines have been exclusively focused on prM and E proteins which rely on generating high levels of neutralizing Ab. However, it has been reported that anti-flavivirus Abs are cross-reactive with other species of flavivirus which may facilitate ZIKV infection through the antibody dependent enhancement (ADE) mechanism[62–64]. Since NS1 is absent in the virion, Ab against NS1 will not neutralize ZIKV, but it may provide an alternative antigen for vaccine development because it avoids the potential risk of disease enhancement[65–67].

We also found that NS1 played an important role in modulating ZIKV-specific immune responses. The expression of E protein from prM-E-NS1 constructs (both rVSV and rVSV-G1670A vector) was delayed compared to the prM-E constructs, which was probably due to the difference in size of the polyprotein and proteolytic cleavage of the polyprotein. Importantly, rVSV-G1670A-prM-E-NS1 induced significantly higher Th2 cytokines ($CD4^+IL-5^+$ and $CD4^+IL-10^+$) and Th17 response ($CD4^+IL-17A^+$) than rVSV-G1670A-prM-E ($P < 0.05$, t-test). This suggests that co-expression of NS1 regulated ZIKV-specific T cell immune response which is critical for virus clearance[68,69]. Another advantage of rVSV-G1670A-prM-E-NS1 is that it triggered a high level of NS1 antibody which may contribute to protection, perhaps through Fc receptor dependent mechanism. In fact, the viremia in rVSV-G1670A-prM-E-NS1 was statistically lower than rVSV-G1670A-prM-E ($P = 0.0236$, t-test) at day 3 post-challenge in A129 mice (Fig. 7a), suggesting that rVSV-G1670A-prM-E-NS1 may have a greater protective efficacy. Either the higher T cell immune response or the NS1 Ab could contribute to the enhanced protection against ZIKV infection.

One of the major concerns of VSV-based vaccines is safety, particularly since VSV is neurotropic. In this study, we utilized MTase as a novel approach to attenuate VSV, since mRNA cap methylation will diminish viral protein translation which in turn would diminish viral replication[45,46,70]. We chose rVSV carrying a single point mutation in the SAM binding site (rVSV-G1670A) as a backbone which was specifically defective in G-N-7 but not 2′-O methylation[45]. If necessary, we can choose mutations (such as K1651A and D1762A) in MTase active site lacking both G-N-7 and 2′-O methylation, which will further enhance the safety of the VSV vector[46,70]. We showed that recombinant rVSV-G1670A-prM-E-NS1 and rVSV-G1670A-prM-E were significantly more attenuated in mice than wild type expressing the same ZIKV genes. Particularly, a dose of $10^5$ PFU of rVSV-G1670A-prM-E-NS1 was completely attenuated in A129 mice whereas rVSV-prM-E-NS1 still caused mortality. In addition, we found that co-expression of NS1 with prM-E offered the additional benefit of further VSV attenuation. In all cases, prM-E-NS1 constructs were significantly more attenuated than prM-E constructs in mice.

In summary, we have developed a novel mtdVSV-based vaccine co-expressing prM, E, and NS1 proteins which is safe and highly efficacious in mice models. This study also highlights a major gap in our understanding of the roles of NS1 protein in regulating ZIKV immune response and protection.

## Methods

**Cell lines, viruses, and plasmid construction.** BHK-21 cells (ATCC no. CCL-10), Vero (ATCC no. CCL-81), and 293T cells (ATCC no. CRL-3216) were purchased from American Type Culture Collection (ATCC, Manassas, VA). BSRT7 cells (kindly provided by Sean Whelan), which stably express T7 RNA polymerase, are clones of BHK-21 cells. All cell lines were grown in Dulbecco's modified Eagle's

medium (DMEM; Life Technologies) supplemented with 10% FBS. ZIKV Cambodian strain (FSS13025) was provided by the World Reference Center for Emerging Viruses and Arboviruses (University of Texas Medical Branch), propagated in Vero cells, and titrated using a standard plaque assay[71]. Plasmids encoding VSV N (pN), P (pP), and L (pL) genes, and an infectious cDNA clone of the viral genome, pVSV1(+), were generous gifts from Gail Wertz[72]. Plasmid pVSV1(+) GxxL, which contains SmaI and XhoI at the G and L gene junction, was kindly provided by Sean Whelan. The full-length envelope (E) gene (from amino acids 1–504) and E truncation mutants [E404 (amino acids 1–404); E414 (amino acid 1-414), and E415 (amino acid 1–415)] lacking the predicted stem-transmembrane domain (TM) were amplified from an infectious cDNA clone of ZIKV Cambodian strain (GenBank accession no. MH158236) by high fidelity PCR. These DNA fragments were digested with SmaI and XhoI and cloned into pVSV(+)GxxL at the same sites. The resulting plasmids were designated pVSV(+)-E, pVSV(+)-E404, pVSV(+)-E414, and pVSV(+)-E415. Using the same strategy, the anchor C (signal peptide) with E, E404, E414, and E415 were cloned into pVSV(+)GxxL at SmaI and XhoI sites resulted in construction of pVSV(+)-aE, pVSV(+)-aE404, pVSV (+)-aE414, and pVSV(+)-aE415 respectively. In addition, the anchor C-premembrane-envelope (prM-E), and anchor C-prM-E truncation mutants (prM-E404, prM-E414, and prM-E415), and anchor C-premembrane-envelope-nonstructural protein 1 (prM-E-NS1) genes were cloned into pVSV(+)GxxL at SmaI and XhoI sites. The resulting plasmids were designated pVSV(+)-prM-E, pVSV(+)-prM-E404, pVSV(+)-prM-E414, pVSV(+)-prM-E,415, and pVSV (+)-prM-E-NS1. Similarly, the anchor C-NS1 gene (amino acids 1-352) was cloned into pVSV(+)GxxL at the SmaI and XhoI sites, and the resultant plasmid was named pVSV(+)-NS1. To further attenuate the VSV vector, we took advantage of a point mutation, G1670A, in the large (L) polymerase protein, which rendered a recombinant virus that is specifically defective in mRNA cap G-N-7, but not 2′-O methylation. Using site-directed mutagenesis, G1670A mutation was introduced into pVSV(+)-E, pVSV(+)-aE, pVSV(+)-prM-E, pVSV(+)-prM-E-NS1, and pVSV(+)-NS1 which resulted in the construction of pVSV(+)-G1670A-E, pVSV (+)-G1670A-aE, pVSV(+)-G1670A-prM-E, pVSV(+)-G1670A-prM-E-NS1, and pVSV(+)-G1670A-NS1 respectively. To prepare DNA vaccine plasmids, the anchor C-prM-E and anchor C-NS1 genes were cloned into pCI vector (Promega) which resulted in the construction of pCI-prM-E and pCI-NS1 respectively. All of the constructs were confirmed by sequencing.

**Recovery of recombinant VSV expressing ZIKV antigens.** Recovery of recombinant VSV (rVSV) from the infectious clone was carried out as described previously[72]. Briefly, rVSV was recovered by cotransfection of plasmid encoding VSV genome, and support plasmids encoding VSV nucleocapsid complex (pN, pP, and pL) into BSRT7 cells infected with a recombinant vaccinia virus (vTF7-3) expressing T7 RNA polymerase (kindly provided by Dr. Bernard Moss). At 96 h post-transfection, cell culture fluids were collected and filtered through a 0.2-μm filter, and the recombinant virus was further amplified in BSRT7 cells. Subsequently, the viruses were plaque purified as described previously[72]. Individual plaques were isolated, and seed stocks were amplified in BSRT7 cells. The viral titer was determined by a plaque assay performed in Vero cells.

**RT-PCR.** Viral RNA was extracted from recombinant VSVs by using an RNeasy minikit (Qiagen, Valencia, CA) according to the manufacturer's instructions. ZIKV genes were amplified by a One Step RT-PCR kit (Qiagen) using primers annealing to VSV G gene at position 4524 (5′-CGAGTTGGTATTTATCTTTGC-3′) and L gene at position 4831 (5′-GTACGTCATGCGCTCATCG-3′) (numbering refers to the complete VSV Indiana genome sequence). The amplified products were analyzed on 1% agarose gel electrophoresis and sequenced.

**Single-cycle growth curves.** Confluent BSRT7 cells were infected with individual viruses at a multiplicity of infection (MOI) of 3. After 1 h of absorption, the inoculum was removed, the cells were washed twice with Dulbecco's modified Eagle's medium (DMEM), fresh DMEM (supplemented with 2% fetal bovine serum) was added, and the infected cells were incubated at 37 °C. Aliquots of the cell culture fluid were removed at the indicated intervals, and virus titers were determined by plaque assay in Vero cells.

**Analysis of the expression of ZIKV antigens by VSV.** Confluent BSRT7 cells were infected with rVSV expressing ZIKV antigens, parental rVSV, or rVSV-G1670A at an MOI of 3.0 as described above. After 3 h post-infection, cells were washed with methionine- and cysteine-free (M⁻ C⁻) medium and incubated with fresh M⁻ C⁻ medium supplemented with actinomycin D (15 μg/ml). After 1 h of incubation, the medium was replaced with M⁻ C⁻ medium supplemented with EasyTag 35S-Express (4 μCi/ml; Perkin-Elmer, Wellesley, MA). After 4 h of incubation, cytoplasmic extracts were prepared and analyzed by sodium dodecyl sulfate-polyacrylamide gel electrophoresis (SDS-PAGE) as described previously[70]. Labeled proteins were detected using a phosphorimager. The original uncropped images of SDS-PAGE are shown in Supplementary Figure 21.

**Detection of ZIKV antigen by western blot.** BSRT7 cells were infected with each rVSV expressing ZIKV antigen as described above. For DNA vaccine plasmid,

HEK293T cells were transfected with pCI, pCI-prM-E, or pCI-NS1 using lipofectamine 2000. At the indicated times post-infection, cell culture medium was harvested and clarified at 4500 × g for 15 min and further concentrated at 35,000 × g for 1.5 h. In the meantime, cells were lysed in lysis buffer containing 5% β-mercaptoethanol, 0.01% NP-40, and 2% SDS. Proteins were separated by 12% SDS-PAGE and transferred to a Hybond enhanced chemiluminescence nitrocellulose membrane (Amersham) in a Mini Trans-Blot electrophoretic transfer cell (Bio-Rad). The blot was probed with rabbit anti-ZIKV E (Cat. No. ZEND20-A, Alpha diagnostic Intl Inc., San Antonio, TX) or NS1 antibody (Cat. No. ZNS111-S, Alpha diagnostic Intl Inc.) at a dilution of 1:2000, followed by horseradish peroxidase-conjugated goat anti-rabbit IgG secondary antibody (Cat. No. sc-2030, Santa Cruz) at a dilution of 1:5000. The blot was developed with SuperSignal West Pico chemiluminescent substrate (Thermo Scientific) and exposed to Kodak BioMax MR film. All original uncropped images of western blot are shown in Supplementary Figs. 16–23.

**Production and purification of ZIKV VLPs by a VSV vector.** Recombinant rVSV-E, rVSV-prM-E, or rVSV-prM-E-NS1 was inoculated into 10 confluent T150 flasks of BSRT7 cells at an MOI of 0.01 in a volume of 2 ml of DMEM. At 1 h postabsorption, 20 ml of DMEM (supplemented with 2% fetal bovine serum) was added to the cultures, and infected cells were incubated at 37 °C for 24–48 h. Cell culture fluids were harvested when extensive cytopathic effect (CPE) was observed. Cell culture fluids were clarified by centrifugation at 3000×g for 30 min. Virus was concentrated through a 40% (wt/vol) sucrose cushion by centrifugation at 30,000×g for 2 h at 4 °C in a Ty 50.2 rotor (Beckman). The pellet was resuspended in NTE buffer (100 mM NaCl, 10 mM Tris, 1 mM EDTA [pH 7.4]) and further purified through a CsCl isopycnic gradient by centrifugation at 35,000×g for 18 h at 4 °C in an SW55 rotor (Beckman). The final pellet was resuspended in 0.3 ml of NTE buffer. Purified ZIKV VLPs were analyzed by SDS-PAGE, western blotting, and electron microscopy. The protein concentrations of the VLPs were measured by using the Bradford reagent (Sigma Chemical Co., St. Louis, MO).

**Purification of ZIKV.** Ten confluent T150 flasks of Vero cells were infected with ZIKV Cambodian strain at an MOI of 0.01 in a volume of 2 ml of DMEM. After 1 h of absorption, 20 ml of DMEM (supplemented with 5% fetal bovine serum) was added, and infected cells were incubated at 37 °C for 72 h. When extensive cytopathic effect (CPE) was observed, cell culture fluids were harvested for ZIKV purification, which was the same procedure as VLP purification mentioned above.

**Transmission electron microscopy.** Negative-staining electron microscopy of purified VLPs and ZIKV was performed as described previously. Briefly, 20 μl of VLP or ZIKV suspension was fixed in copper grids (Electron Microscopy Sciences, Inc.) and negatively stained with 1% ammonium molybdate. Virus particles were visualized by using a FEI Tecnai G2 Spirit transmission electron microscope (TEM) at 80 kV at the Microscopy and Imaging Facility at The Ohio State University. Images were captured on a MegaView III side-mounted charge-coupled-device camera (Soft Imaging System, Lakewood, CO), and figures were processed using Adobe Photoshop software (Adobe Systems, San Jose, CA).

**Animal experiments.** All mice were housed within ULAR facilities of The Ohio State University under approved Institutional Laboratory Animal Care and Use Committee (IACUC) guidelines (protocol no. 2009A0160). Each inoculation group was separately housed in rodent cages under biosafety level 2 (BSL-2) conditions.

Experiment 1: determine whether VSV constructs are immunogenic in BALB/c mice. Sixty 4 to 6-week-old specific-pathogen-free female BALB/c mice (Charles River Laboratories, Wilmington, MA) were randomly divided into 12 groups (5 mice per group). Mice in group 1 were inoculated with parental rVSV (with no insertion). Mice in groups 2–10 were inoculated with nine different rVSVs expressing ZIKV antigens (rVSV-E404, rVSV-E414, rVSV-E415, rVSV-E, rVSV-prM-E404, rVSV-prM-E414, rVSV-prM-E415, rVSV-prM-E, and rVSV-prM-E-NS1). Mice in group 11 were inoculated with DMEM and served as uninfected controls (the normal control). Mice in group 12 were immunized with DNA vaccine pCI-prM-E. For VSV, each mouse was inoculated intranasally at a dose of $1 \times 10^6$ PFU in a volume of 50 μl. For DNA vaccine, mice were immunized intramuscularly with 50 μg of pCI-prM-E, and boosted with same dose two weeks later. After inoculation, the animals were evaluated twice every day for mortality and the presence of any symptoms of VSV infection. The severity of clinical signs associated with VSV infection was scored based on the following criteria: grade 3 (severe) was characterized by ruffled fur, hyperexcitability, tremors, circling, and paralysis; grade 2 (moderate) was characterized by ruffled fur with neurological symptoms such as circling; grade 1 (mild) was characterized by ruffled fur but no neurological symptoms; grade 0 was defined as no symptoms. The body weight of each mouse was monitored every three days. Blood samples were collected from each mouse weekly by bleeding facial vein, and serum was isolated for antibody detection. At week 5 post-inoculation, all mice were euthanized.

Experiment 2: determine antibody and T cell immune responses triggered by mtdVSV in BALB/c mice. Thirty 6-week-old SPF female BALB/c mice (Charles River Laboratories) were randomly divided into six groups (5 mice per group). Mice in group 1 were inoculated with DMEM and served as unimmunized controls

(normal control). Mice in groups 2–6 were immunized with rVSV, rVSV-G1670A, rVSV-G1670A-E, rVSV-G1670A-prM-E, and rVSV-G1670A-prM-E-NS1. All mice were immunized intranasally at a dose of $1 \times 10^6$ PFU per mouse. After immunization, the animals were evaluated daily for body weight, mortality, and the presence of any symptoms of VSV infection. Blood samples were collected from each mouse weekly by bleeding facial vein, and serum was isolated for antibody detection. At week 5 post-inoculation, all mice were euthanized, and whole blood and spleens were isolated from each mouse for a T cell assay.

Experiment 3: determine whether mtdVSV vaccine can protect BALB/c mice against viremia until day 24 after ZIKV challenge. Seventy 4-week-old SPF BALB/c mice (Charles River Laboratories) were randomly divided into 7 groups (10 per group, 5 female and 5 male). Mice in groups 1–5 were immunized with DMEM, rVSV-G1670A, rVSV-G1670A-E, rVSV-G1670A-prM-E, or rVSV-G1670A-prM-E-NS1. Mice in group 6 were immunized with DNA vaccine. Mice in group 7 were served as normal control (immunized with DMEM and unchallenged). For VSV, mice were inoculated intranasally at a dose of $1 \times 10^6$ PFU per mouse. For DNA vaccine, mice were immunized intramuscularly with 50 μg of pCI-prM-E, and boosted with same dose 2 weeks later. After immunization, the presence of any VSV symptom induced by mtdVSV-based ZIKV vaccine candidates was evaluated twice a day. At week 5 post-immunization, mice in groups 1–6 were challenged intravenously with ZIKV Cambodian strain at a dose of $1 \times 10^6$ PFU per mouse. At 24 h prior to ZIKV challenge, mice were intraperitoneally administered 1.8 mg of anti-IFNAR1 (Leinco Technologies, Fenton, MO) blocking antibody. After challenge, the animals were evaluated twice daily for mortality and the presence of any symptoms of ZIKV infection. The body weight for each mouse was monitored daily. At day 24 post-challenge, all mice from each group were euthanized. The blood, brain, lungs, liver, and spleen from each mouse were collected for virus quantification and histologic evaluation.

Experiment 4: determine whether mtdVSV vaccine can protect BALB/c mice against viremia until day 7 post-challenge. Mice (6-week-old) in groups 1–5 were immunized with saline, rVSV-G1670A, rVSV-G1670A-aE, rVSV-G1670A-prM-E, or rVSV-G1670A-prM-E-NS1. The mice in group 6 served as a normal control (unimmunized unchallenged). The experimental procedure was identical to Experiment 3 except the mice were euthanized at day 7 after challenge with ZIKV Cambodian strain.

Experiment 5: determine whether mtdVSV vaccine and DNA vaccine can protect A129 mice against ZIKV challenge. Thirty five 6-week-old female A129 mice (Jackson Laboratories) were randomly divided into 7 groups (5 per group). Mice in groups 1–3 were immunized intramuscularly with pCI, pCI-prM-E, or pCI-NS1 at a dose of 50 μg DNA per mouse. Two weeks later, mice in groups 1–3 were boosted intramuscularly with the same plasmid at the same dose. Mice in groups 4–6 were administered intramuscularly using a single dose ($1 \times 10^5$ PFU per mouse) of rVSV-G1670A-prM-E-NS1, rVSV-prM-E-NS1, or rVSV-prM-E. Mice in group 7 were served as unvaccinated unchallenged control. After immunization, mice were evaluated every three days for body weight. The safety of mtdVSV-based ZIKV vaccine candidates was evaluated twice a day. Blood samples were collected at week 1 and 3 from each mouse for detection of antibody. At week 4 post-immunization, mice in groups 1–6 were intraperitoneally challenged with ZIKV Cambodian strain at a dose of $1 \times 10^5$ PFU per mouse. After challenge, the animals were evaluated twice every day for mortality and the presence of any symptoms of ZIKV infection. The severity of clinical disease was scored based the following criteria: 1 = heathy; 2 = mild; 3 = moderate; and 4 = severe, and early removal is required. The body weight for each mouse was monitored daily. Blood was collected at days 3 and 7 for the detection of viremia. At day 7 post-challenge, all mice from each group were euthanized, and brain, lungs, uterus/ovary, and spleen from each mouse were collected for virus quantification and histologic evaluation.

Experiment 6: determine whether NS1 alone can protect BALB/c mice against viremia. Twenty five 4-week-old female BALB/c mice were randomly divided into 5 groups (5 per group). Mice in groups 1–5 were immunized with DMEM, pCI, pCI-NS1, rVSV-G1670A-NS1, or rVSV-G1670A-prM-E. For VSV, mice were inoculated intranasally at a dose of $1 \times 10^6$ PFU per mouse. For DNA vaccine, mice were immunized intramuscularly with 50 μg of plasmid, and boosted with same dose two weeks later. At week 4 post-immunization, mice in groups 2–6 were intravenously challenged with ZIKV Cambodian strain at a dose of $1 \times 10^6$ PFU per mouse. At 24 h prior to ZIKV challenge, mice were intraperitoneally administered 1.8 mg of anti-IFNAR1 (Leinco Technologies) blocking antibody. At days 3 and 7 post-challenge, blood was collected from each mouse for detection of viremia by real-time RT-PCR.

Experiment 7: validate the safety and immunogenicity of rVSV-G1670A-prM-E-NS1 in A129 mice. There were three groups in this study. Mice in group 1 were immunized intramuscularly with a single dose ($1 \times 10^5$ PFU per mouse) of rVSV-G1670A-prM-E-NS1. Mice in groups 2 and 3 were served as unimmunized challenged control and normal control (unimmunized unchallenged). Blood was collected from each mouse weekly for antibody detection. At week 4, mice in groups 1 and 2 were intraperitoneally challenged with ZIKV Cambodian strain at a dose of $1 \times 10^5$ PFU per mouse. After challenge, mice were monitored for body weight changes and viremia every 1 or 3 days for 21 days.

**Detection of ZIKV E or NS1-specific antibody by ELISA.** Ninety-six-well plates were first coated with 50 μl of highly purified ZIKV E or NS1 protein

(MyBioSource, Inc., San Diego, CA) (4 µg/ml, in 50 mM Na2CO3 buffer, pH 9.6) per well at 4 °C overnight, and then blocked with Bovine Serum Albumin (BSA, 1% W/V in PBS, 100 µl/well) at 37 °C for 2 h. Subsequently, individual serum samples were tested for ZIKV-specific Ab on antigen-coated plates. Briefly, serum samples were 2-fold serially diluted and added to E or NS1 protein-coated wells. After incubation at room temperature for 2 h, the plates were washed five times with phosphate-buffered saline (PBS)-Tween (0.05%), followed by incubation with 50 µl of goat anti-mouse IgG horseradish peroxidase (HRP)-conjugated secondary Abs (Sigma) at a dilution of 1:2000 for 1 h. Plates were washed, developed with 100 µl of 3,3′,5,5′-tetramethylbenzidine (TMB), stopped by 100 µl of $H_2SO_4$ (2 mol/L), and the optical density (OD) at 450 nm was determined by BioTek microplate reader. Endpoint titers were determined as the reciprocal of the highest dilution that had an absorbance value 2.1-folds greater than the background level (DMEM control). Ab titers were calculated by the geometric mean titers (GMT).

**Detection of ZIKV neutralizing Ab**. ZIKV-specific neutralizing Ab was determined using a microneutralization (MN) assay as described with modifications[17]. Briefly, serum samples were serially diluted twofold in 96-well micro-plates, and 100 µl of virus solution containing 50 PFU of ZIKV-Cambodian strain was added to 100 µl of each serum dilution and incubated at 37 °C for 1 h. The mixtures were then transferred to 24-well plates containing confluent Vero cell monolayers. After incubation for 60 h at 37 °C, cells were fixed with 4% (vol/vol) phosphate-buffered paraformaldehyde for 1 h and washed three times with PBS. After permeation by 0.4% (vol/vol) Triton-X 100 at room temperature for 20 min and washed for three times, a ZIKV E monoclonal antibody (MyBioSource, Inc.) was added to each well at a dilution of 1:1000 and incubated at 37 °C for 2 h, followed by washing with PBS three times. A horseradish peroxidase-conjugated goat anti-rabbit IgG secondary antibody (Santa Cruz) at a dilution of 1:2000 was added to each well and incubated at 37 °C for 1 h, followed by washing with PBS for three times. The plates were then developed with 3-Amino-9-ethylcarbazole (AEC) substrate for 1 h at room temperature and stained plaques in each well were counted under light microscope. The half maximal inhibitory concentration ($IC_{50}$) of neutralizing antibody in mice serum was calculated based on the number of plaques in each well compared with the average value of DMEM group.

**Analysis of ZIKV-specific T cell responses**. To determine the nature of T cell responses that supported the development of ZIKV-specific Ab responses by rVSV expressing ZIKV antigens, we analyzed cytokine production by ZIKV E-specific spleen T cells. More specifically, spleen cells were aseptically removed from mice 35 days after immunization, and minced by pressing through a cell strainer. Red blood cells were removed by incubation in 0.84 % ammonium chloride and, following a series of washes in RPMI 1640, cells were resuspended in RPMI 1640 supplemented with 2 mM L-glutamine, 1 mM sodium pyruvate, 10 mM HEPES, 100 U/ml penicillin, 100 µg/ml streptomycin, and 10% fetal calf serum. The cell concentrations were adjusted to $3 \times 10^6$ cells/mL and 100 µl were added into each well (3 wells per spleen sample) of a 96-well microtiter plate and cultured either alone or in the presence of 20 µg/ml of ZIKV E protein for 5 days at 37 °C in a 5% CO2 atmosphere. Culture supernatants were collected from each well and frozen at −80 °C until analysis of secreted cytokines using the Bio-Plex Pro Mouse Cytokine Standard 23-Plex, Group I (Bio-Rad Laboratories Inc, Hercules, CA) per manufacturer's instructions. The frequencies of ZIKV-specific Th1 (IFN-α+CD4+CD3+ and TNF-β+CD4+CD3+), Th2 cells (IL-4+CD4+CD3+, IL-5+CD4+CD3+), Th17 (IL-17A+ CD4+ CD3+), and Tfh (IL-21+ CD4+ CD3+) cells were determined by intracellular staining with the corresponding anti-cytokine Abs (at a dilution of 1:5000) after additional incubation in the presence of PMA and ionomycin. Cytokine-specific antibodies including Alexa Fluor 700 anti-CD3 (Cat. No. 100216), Alexa Fluor 750 anti-CD4 (Cat. No. 100460), Alexa Fluor 488 anti- IFNγ (Cat. No. 505813), PerCP Cy5.5 anti-TNFα (Cat. No. 506322), PE anti-IL-5 (Cat. No. 504307), Alexa Fluor 647 anti-IL-21 (Cat. No. 516803), PECy7 anti-IL-10 (Cat. No. 505026), Brilliant Violet 650 anti-IL-17 (Cat. No. 506929), Brilliant Violet 605 anti-IL-4 (Cat. No. 504125) were purchased from Biolegend (San Diego, CA). The cells were then analyzed with the aid of an Attune flow cytometer and data were expressed as mean % positive cells ± standard deviation.

**Measurement of viral burden**. At indicated time points after ZIKV challenge, blood was collected and organs were recovered. Organs were weighed and homogenized using a bead-beater apparatus (MagNA Lyser, Roche). The total RNA was extracted from tissue samples and blood by using TRIzol Reagent (Life technologies, Carlsbad, CA). Reverse transcription (RT) was conducted using a primer (5′-CTCGTCTCTTCTTCTCCTTCCTAGCATTGA-3′) targeting the capsid (C) gene of ZIKV and the Superscript III transcriptase kit (Invitrogen, Carlsbad, CA). The RT products were then used to perform real-time PCR using primers specifically targeting the E gene of ZIKV (forward, 5′-CATCAGGATGGTCT TGGCGATTCTAGC-3′; reverse, 5′-CTCGTCTCTTCTTCTCCTTCCTAGCATT GA-3′) in a StepOne real-time PCR system (Applied Biosystems). A standard curve was generated using a serial dilution of ZIKV RNA extracted from known PFU titer of infectious virus. Amplification cycles used were 2 min at 50 °C, 10 min at 95 °C, and 40 cycles of 15 s at 95 °C and 1 min at 60 °C. The threshold for detection of fluorescence above the background was set within the exponential phase of the amplification curves. For each assay, 10-fold dilutions of standard viral RNA were generated, and negative-control samples and double-distilled water (ddH2O) were included in each assay. Viral burden is expressed on a log10 scale as viral RNA equivalents per gram or per milliliter. Similarly, total RNA from brain tissues was used for quantification of VSV RNA using primers annealing to the VSV L gene and shown as RNA copies per gram, using a plasmid encoding VSV genome as a standard curve.

**Histology**. Half of the tissues (brain, lung, uterus/ovary, and spleen) from each experiment were preserved in 4% (vol/vol) phosphate-buffered paraformaldehyde. Fixed tissues were embedded in paraffin, sectioned at 5 µm, and stained with hematoxylin-eosin (HE) for the examination of histological changes by light microscopy. The pathologist was blinded to each group.

**Quantitative and statistical analyses**. Quantitative analysis was performed either by densitometric scanning of autoradiographs or by using a phosphorimager (Typhoon; GE Healthcare) and ImageQuant TL software (GE Healthcare, Piscataway, NJ). All experiments were three replicates. No animals or samples were excluded from the analysis. Data were collected and assessed blindly for each group allocation during the experiment. Statistical analysis was performed by one-way multiple comparisons using SPSS 8.0 statistical analysis software (SPSS Inc., Chicago, IL). A P value of ≤ 0.05 was considered statistically significant. All materials, reagents, and data reported in this manuscript are available from the authors upon request.

**Data availability**. The authors declare that the data supporting the findings of this study are available with the article and its Supplementary Information files, or are available from the corresponding author upon request.

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

## Acknowledgements

This work was supported by internal funds from The Ohio State University (Jianrong Li and Prosper Boyaka) and Nationwide Children Hospital (Mark Peeples), and a grant from the NIH-NIAID (P01AI112524 supporting Mark Peeples and Jianrong Li). Work in

Shan-Lu Liu's lab was supported by NIH grants R01AI112381 and R21AI109464. P.-Y.S. lab was supported by University of Texas STARs Award, CDC grant for the Western Gulf Center of Excellence for Vector-Borne Diseases, Pan American Health Organization grant SCON2016-01353, the Kleberg Foundation Award, UTMB CTSA UL1TR-001439, and NIH grant AI127744. We thank Gail Wertz for the VSV reverse genetics system and Sean Whelan for providing pVSV(+) GxxL plasmid. We thank Edward Calomeni at The Ohio State University for electron microscopy analysis.

## Author contributions

A.L., Y.M., M.E.P., and J.L. conceived the concept of using MTase-defective VSV-based ZIKV vaccine strategy. A.L., J.Y., M.L., Y.M., Z.A., C.S., M.X., X.L., K.C., N.M., J.H., R.J., P.Y.S., M.E.P., S.L.L., P.N.B., and J. L. performed experiments and data analysis. A.L., J.Y., M.L., Y.M., R.J., P.Y.S., M.E.P., S.L.L., P.N.B., and J.L. designed the experiments and interpreted the results. A.L., M.L., R.J., P.Y.S., M.E.P., S.L.L., P.N.B., and J.L. wrote the manuscript, and all other authors edited the manuscript.

## Additional information

**Competing interests:** The authors declare no competing interests.

