## [Peer Review File · Nature Communications]

Reviewers' comments:

Reviewer #1 (Remarks to the Author):

This study presents a series of experiments in BALB/c and A129 mice investigating the immunogenicity and protective capacity of various rVSV-vectored and DNA ZIKV vaccine candidates. The rVSV candidates include those that express ZIKV prM-E-NS1, prM-E, or E alone. The E and prM-E constructs were each further modified to include 3' E truncations of the TM domain, generating soluble E. Finally, further attenuation was achieved through a point mutation in the VSV large polymerase (G1670A). The main emphasis is on the rVSV-G1670A-prM-E-NS1 candidate, with the hypothesis that NS1 might provide added protection to the prM-E structural genes that are already the basis of many flavivirus (and ZIKV) vaccines for their ability to produce subviral particles that elicit E-specific neutralizing antibodies. This hypothesis is supported in the current study by the novel finding of partial protection from ZIKV infection after vaccination with a DNA vaccine expressing only ZIKV NS1.

Two major comments:

1. The antigen released from VSV-infected or DNA transfected cells (VLPs, E alone, NS1 alone) should be better characterized, as this is the relevant immunogen. Western blots and/or ELISAs using ZIKV specific reagents should be used to characterize the supernatants. Although the methods section indicates that Western blot analysis was performed with cell supernatants, all Western blots presented (Figures 2, 3, S2) include only cell lysates. This is particularly relevant for the E only vaccine candidates. Prior studies of other flaviviruses indicate a critical role for prM in proper folding and release of the E protein in mammalian cells (shown for TBEV in Allison et al., PMID: 10364309 and 7637027). Thus, it is perhaps not surprising that the ZIKV E only constructs were not effective immunogens as they could be improperly folded/ not released from cells. The inclusion of the E only vaccine candidates could potentially be removed altogether.
2. The sequences of the constructs need to be detailed in the methods, particularly the amino acids at the start and end of prM, E, and NS1, and information about the leader sequence used prior to prM and E. It is surprising that the prM-E DNA vaccine in this study resulted in such a poor immune response, when a seemingly identical vaccine elicited robust binding and neutralizing antibody responses by 2 weeks post-vaccination with only a single dose, also in BALB/c mice (Dowd, et al. PMID 26530385, supplementary material). This discrepancy should be addressed and the amino acid sequences clarified in the manuscript.

The inclusion of NS1 in a prM-E subviral particle vaccine is relevant to the field, as recent studies have indicated that anti-NS1 antibodies can be partially protective against flavivirus infection. However, the vaccine constructs need to be described and characterized in greater detail prior to making comparisons and drawing conclusions regarding their immunogenicity.

Additional comments:

The majority of figures need clarification of the number of repeats performed, what statistics are shown (i.e. median versus mean), and error bars should be included and clearly denoted in the figure legend. Experiments from which a difference in growth kinetics, etc. are concluded should be confirmed in multiple independent experiments (for example attenuation of rVSV-G1670A-prM-E-NS1 in Figure 4B).

The authors should include the following citation in their discussion of the protective effect of NS1 antibodies (Beatty, et al. PMID 26355030).

Line 189- states that rVSV-prM-E-NS1 is attenuated in growth, but Figure S1 (lines 172-174) states that all recombinant viruses had similar growth kinetics.

Line 200- Figure S2, not S2A

Lines 206-208- Figure 2 would benefit from arrows indicating VLPs, and panel 2E discriminating VLPs versus VSV

Figure 1a and 4b- Images of plaques are not helpful. The size description in the text provides enough context.

Figure 5a- What are the statistics to support lines 284-286 that no weight loss occurred? G1670A-prM-E looks to have a slight dip at day7 (also seen in Figure S4).

Figure 6- Group 3 mislabeled, should be G1670-A-E.

Figure 7A- The viremia data should be shown for all groups, not just the control group.

Reviewer #2 (Remarks to the Author):

In the manuscript entitled "A novel Zika virus vaccine expressing pre-membrane, envelope, and NS1 proteins" authors investigated the viability of vesicular stomatitis virus (VSV) expressing the ZIKAV full length envelope protein or truncated proteins, isolated or together with prM and NS1 proteins, as a vaccine strategy against Zika. The immune response and protection induced by these viruses was analyzed in mice. The manuscript is well written and results are consistent. However there are some points that could improve the quality of the manuscript, that will be listed below.

Introduction:

Pg 3: Authors should use the term "Congenital Zika Syndrome" which has been preferable adopted for description of the effects of Zika infection in the fetus and infants.

Pg 6: Authors cited another paper that also tested the VSV expressing prM-E from ZIKAV in mice. They should discuss this article with more detail in the discussion section presenting what was different from the present manuscript.

Results:

Pg 7: The presentation of the Zika envelope protein structure is very poor. Authors should explain that it is composed of an ectodomain, consisting of three domains (I, II and III), and stem and transmembrane regions. They could cite the reference of Sirohi et al, Science 2016, which investigated the structure of the virus. Such explanation could perhaps be inserted in the introduction section and in results or materials and methods authors could indicate which part was deleted in truncate E proteins. Did they remove all stem and transmembrane regions?

In fig. S2 one band corresponding to the NS1 protein should also not be present in rVSV-prM-E-NS1 infected cells?

Pg 11: In the last sentence authors affirmed that results demonstrated that inoculation with rVSV-prM-E-NS1 induced high antibody levels 1 or 2 weeks after inoculation, but this is not what we observe in fig 3. As we see in fig 3, antibody levels in this group were similar to those detected in mice immunized with truncated E protein without prM.

Pg 15: Authors affirmed that results demonstrated that co-expression of NS1 enhances Th2 and Th17 responses but they should also comment that in this group the IFN-g was also higher comparing to the other groups (fig 6B), suggesting a Th1 response. It seems that the presence of NS1 lead to a more balanced response including both Th1 and Th2 cells.

Pg15/16: Authors affirmed that the mtdVSV-based vaccines were safe, but, as we saw in fig S4, animals immunized with rVSV-G1670A-prM-E also lost body weight. Overall, they should be more cautious in asserting about the safety of these viruses based on the present study.

Pg 17: Why did authors immunized A129 animals by the intramuscular route while BALB/c mice were inoculated intranasally? Is it because A129 are more susceptible and the intranasal route would lead to high morbidity and death?

Standard deviations should be inserted in several graphs (Figs 3, 5A, S1, S3, S4, S5), informing it in figure legends.

Discussion:

Pg 21: Authors compared the two immunization routes (intramuscular and intranasal) and suggested that it may play a role in the antibody response, by delaying it, for instance. However, in this case it was not only the inoculation route that could influence results but also the background of each animal (BALB/c and A129).

Pg 22: Authors compared the immune response induced by the DNA and VSV-based vaccines. It should also be interesting to evaluate the long-term response induced by both strategies.

Reviewer #3 (Remarks to the Author):

Review: A novel Zika virus vaccine expressing pre-membrane, envelope, and NS1 proteins.

The manuscript describes the generation and characterization of VSV based Zika vaccines. Anzhong Li et.al have convincingly demonstrated protection against Zika challenge using their MTase defective VSV-prME-NS1, which is less pathogenic than the wild-type virus. However, there are some weaknesses. First of all, there are way too many figures – this should be restricted to a maximum of 6 or 7 – a lot of the results can be consolidated and

some of the results can be removed, for example the NS1 partial protection statement, which is not convincing and based on 1! mouse.

Major comments and concerns:

Line 49-51: Based on current published data, neutralizing antibodies to the Zika envelope is essential for viral clearance. More importantly, a Zika vaccine without the envelope protein but containing the NS1 antigen only, has not been shown to protect against Zika challenge in this manuscript. This sentence needs to be removed from the abstract.

Line 218,282,336: An intranasal route of immunization is not ideal, as it cannot be translated to humans. Why did the authors not do an intra-muscular immunization?

Line 213-214: Zika virus control would be required in order to conclusively draw similarities between the VLP's made by the rVSV-Zika constructs and Zika virus. The EM is of low quality and the conclusion not supported by the presented data. Remove Figure 2E and the VLP statements from the manuscript.

Line 227-233: The authors have conducted pathogenicity experiments (intranasal inoculations) with their recombinant VSV-Zika vaccines and used weight loss as a measure of pathogenicity. Looking at figure S3, most mice except for the pCI-prME and DMEM controls show varying degree of weight loss, indicating that the constructs do show some pathogenicity at earlier time points. In order to conclusively prove that the recombinant vaccines are truly non-pathogenic, mouse brains should have been harvested at endpoints and analyzed by for viral VSV RNA.

Figure 4 (A,B,C): Important controls - rVSV-prME and rVSV-prME-NS1 and rVSV are not in all the figures. Ideally these attenuated rVSV's should express less of the Zika E in comparison to the rVSV-prME.

Line 337: The dose and route of Zika challenge should be mentioned in the result, legend and the method section. The reader has to go through result, figure legend and method to get the complete information!

Line 361: It is important to show the complete clearance of Zika virus on day 7 and future time points for the vaccinated group and unvaccinated group. Why is this data not shown? The Zika viral RNA data for different groups and different days could be consolidated into one graph replacing figure 7A and B.

Figure 9A+9B can be consolidated into 1 graph i.e. week 1 and 3, Zika E IgG Response. Also mention in the legend that the titers shown are total IgG titers.

Figure 9C+9D can be consolidated into 1 graph.

Figure 9E and F can also be consolidated into 1 graph.

Line 387: The strain and route of Zika virus challenge should be mentioned. The reader has to go through the result, figure legend and method to get the complete information. Confusing!

Line 411-435: The experiment was terminated at day 7 and at this point the mice that were immunized with pCI-NS1 had viremia (in the brain, lung spleen and uterus), symptoms and encephalitis. Hence, it is difficult to conclude partial protection based on slightly reduced symptoms (symptoms are bias criteria in the first place). rVSV-G170A-NS1 construct should have been used to study this. Why did the authors suddenly switch to the DNA vaccine platform to study this?. These data need to be removed, they are off topic.

Additionally, the authors could have included VSV viral load in the brains of the AG129 mice immunized with Mtase defective- rVSV-Zika vaccines and challenged with Zika virus as a supplemental figure to rule out pathogenicity of the viral vectors in these immunocompromised mice.

Line 445: NS1 providing partial protection cannot be concluded as the T cell responses were seen for the rVSV-G1670A-prME-NS1 construct and not for the pCI-NS1 DNA vaccine.

Line 721: Route of ZIKV challenge is not mentioned in the method section.

Minor comments:

Line 200: Re-label to figure S2. S2A doesn't exist.

Figure 3,5,9: legend and figure should mention that the ELISA titers shown are mean endpoint titers of 5 mice.

Figure S2: requires a size marker

Figure S4: The weight's reported in the figure are mean body weights per group and should be mentioned in the figure legend.

Response to reviewers' comments

Introduction: We thank the three reviewers for their careful review of this manuscript. All of them identified a number of constructive and thoughtful comments, which were extremely helpful in revising this manuscript. We have now modified the manuscript according to their comments. Each point is now addressed.

Reviewer #1:

Two major comments:

1. The antigen released from VSV-infected or DNA transfected cells (VLPs, E alone, NS1 alone) should be better characterized, as this is the relevant immunogen. Western blots and/or ELISAs using ZIKV specific reagents should be used to characterize the supernatants. Although the methods section indicates that Western blot analysis was performed with cell supernatants, all Western blots presented (Figures 2, 3, S2) include only cell lysates. This is particularly relevant for the E only vaccine candidates. Prior studies of other flaviviruses indicate a critical role for prM in proper folding and release of the E protein in mammalian cells (shown for TBEV in Allison et al., PMID: 10364309 and 7637027). Thus, it is perhaps not surprising that the ZIKV E only constructs were not effective immunogens as they could be improperly folded/ not released from cells. The inclusion of the E only vaccine candidates could potentially be removed altogether.

Response: Thank you for this comment. We have now provided the Western blot data to characterize the ZIKV antigens released in cell culture supernatants (Please see Fig.S3 and Fig.2D and E). Please see lines 189-195, and lines 288-292. All recombinant VSVs co-expressing prM and E/E truncations released a high level of E/E truncation in supernatants, even without the need for concentration of the supernatants. This suggests that E/E truncations were highly expressed by the VSV vector and were secreted into cell culture medium. In contrast, E/E truncations were undetectable in the supernatants when they were expressed without prM. This is consistent with previous observations that prM plays a critical role in proper folding and release of the E protein for TBEV (Allison et al., PMID: 10364309 and 7637027). These papers have been cited. For our manuscript, we used rVSV-E alone as a control. We showed that E alone was not an effective vaccine candidate when it was tested in mice.

2. The sequences of the constructs need to be detailed in the methods, particularly the amino acids at the start and end of prM, E, and NS1, and information about the leader sequence used prior to prM and E. It is surprising that the prM-E DNA vaccine in this study resulted in such a poor immune response, when a seemingly identical vaccine elicited robust binding and neutralizing antibody responses by 2 weeks post-vaccination with only a single dose, also in BALB/c mice (Dowd, et al. PMID 26530385, supplementary material). This discrepancy should be addressed and the amino acid sequences clarified in the manuscript.

The inclusion of NS1 in a prM-E subviral particle vaccine is relevant to the field, as recent studies have indicated that anti-NS1 antibodies can be partially protective against flavivirus infection. However, the vaccine constructs need to be described and characterized in greater

detail prior to making comparisons and drawing conclusions regarding their immunogenicity.

Response: We have now provided the detailed information of the sequence of each construct in the Materials and Methods. Please see lines 667-670, and Fig.S1.

In our study, we used pCI vector to construct a DNA vaccine (pCI-prM-E) and used it as a control. We showed that the expression of E protein in rVSV-prM-E infected cells was significantly higher than pCI-prM-E transfected cells. In the mice vaccination study, we showed that a single dose vaccination of recombinant rVSV-prM-E triggered significantly higher ZIKV antibody response than pCI-prM-E, despite the fact that mice in pCI-prM-E group have been boosted. Our data suggest that we rVSV-based ZIKV vaccine had a higher immunogenicity than the DNA vaccine.

In our study, we found that the antibody induced at the early time points (weeks 1-3 post-vaccination) by ZIKV DNA vaccine is low. However, at late time points (weeks 4 and 5), we found that the DNA vaccine was capable of triggering high antibodies, consistent with previous studies. The efficacy of DNA vaccine may be affected by many factors such as the plasmid vector, adjuvant, and delivery strategy. Previously, two publications showed that a Zika DNA vaccine encoding prM-E was effective in triggering ZIKV-specific immune response and provided sufficient protection against ZIKV challenge. Larocca et al., (2016) showed that a high titer of ZIKV E-specific ELISA antibody was detected at week 3 when BALB/c mice was immunized intramuscularly with 50 µg of DNA vaccine. However, antibody titer at the early time points (weeks 1 and 2) was not reported in their studies. Dowd et al., (2016) showed that high titer of ZIKV antibody was detected at week 2 in mice when the DNA vaccine was delivered by electroporation. In addition, antibody triggered in C57BL/6 by a DNA vaccine was higher than that in BALB/c mice. We have now discussed these articles in the revised manuscript. Please see lines 547-557.

We thank the reviewer's comment on the role of NS1 in modulating flavivirus immune response. We showed that ZIKV NS1 modulated ZIKV-specific antibody and T cell immune response, and provided the evidence that NS1 alone can provide partial protection against ZIKV infection in A129 mice. As suggested by Reviewer#3, we have now constructed rVSV-G1670A-NS1 and showed that rVSV-G1670A-NS1 provided substantial protection against ZIKV-induced viremia (please see our response to Reviewer #3 for the detail). Previously, NS1 proteins of several of other flaviviruses (such as Dengue virus, West Nile virus, and yellow fever virus) can provide partial or complete protection against the virus challenge. Our results are consistent with the previous observations of NS1 proteins in other flaviviruses.

Additional comments:

3. The majority of figures need clarification of the number of repeats performed, what statistics are shown (i.e. median versus mean), and error bars should be included and clearly denoted in the figure legend. Experiments from which a difference in growth kinetics, etc. are concluded

should be confirmed in multiple independent experiments (for example attenuation of rVSV-G1670A-prM-E-NS1 in Figure 4B).

Response: Thank you for this comment. We have provided this information in the Materials and Methods, and figure legend. The statistics and error bars are indicated in the figures. Data on viral growth kinetics were done in three independent experiments (please see Fig.S2).

4. The authors should include the following citation in their discussion of the protective effect of NS1 antibodies (Beatty, et al. PMID 26355030).

Response: This paper has now been cited and discussed. Please see lines 592-597.

5. Line 189- states that rVSV-prM-E-NS1 is attenuated in growth, but Figure S1 (lines 172-174) states that all recombinant viruses had similar growth kinetics.

Response: Thank you for catching this point. Three independent experiments confirmed that rVSV-prM-E-NS1 did have a significant attenuation in growth compared to rVSV-prM-E, as demonstrated by the plaque size (Fig.S1B) and growth kinetics (Fig.S2). Fig.S1 and Fig.S2 have now been updated.

6. Line 200- Figure S2, not S2A

Response: It is now corrected.

7. Lines 206-208- Figure 2 would benefit from arrows indicating VLPs, and panel 2E discriminating VLPs versus VSV.

Response: Yes, we have now used arrows to indicate the ZIKV VLPs and VSV (Fig.1E and Fig.S5). As suggested by Reviewer#3, we also purified ZIKV virions and performed an EM analysis. An EM image of native ZIKV virion was shown for comparison with ZIKV VLPs. Please see lines 221-224.

8. Figure 1a and 4b- Images of plaques are not helpful. The size description in the text provides enough context.

Response: We agree with the reviewer. We have now reported the images of viral plaques as supplementary figures (Fig.S1B).

9. Figure 5a- What are the statistics to support lines 284-286 that no weight loss occurred? G1670A-prM-E looks to have a slight dip at day7 (also seen in Figure S4).

Response: We have modified these sentences to reflect the data in the Figures. In the Fig.2F, 5 female mice per group were used in the study. Statistical analysis showed that there was no significant difference among DMEM, rVSV-G1670A-prM-E, and rVSV-G1670A-prM-E-NS1 ($P>0.05$). In Fig. S9, 10 mice (5 female and 5 male) per group were used in the study. In Fig.S9, body weight from rVSV-G1670A-prM-E-NS1 had no significant difference with the

DMEM group ($P>0.05$). However, rVSV-G1670A-prM-E had significant less body weight gain at days 3-10 compared to the DMEM control group ($P<0.05$). Mice in rVSV-G1670A-E and rVSV-G1670A-prM-E groups in Fig.S9 had more body weight losses than the same groups in Fig.2F probably because of both female and male mice were used in experiment Fig.S9, whereas only female mice were used in experiment Fig.2F.

In A129 mice (Fig.5A), rVSV-G1670A-prM-E-NS1 did not have significant weight loss ($P>0.05$), whereas rVSV-G1670A-prM-E still experienced significant weight loss from days 3-14 ($P<0.05$).

In combination of Fig.2F, Fig.S9, and Fig.5A, our data showed that rVSV-G1670A-prM-E-NS1 is much more attenuated than rVSV-G1670A-prM-E. We have modified the description of weight changes for each experiment. Please see lines 302-304, lines 361-365, and lines 410-412.

10. Figure 6- Group 3 mislabeled, should be G1670-A-E.

Response: It has been corrected.

11. Figure 7A- The viremia data should be shown for all groups, not just the control group.

Response: Yes, we have now provided viremia data for all groups at day 7 post-challenge (Fig.4C). Please see lines 384-388. For this experiment (Fig.4), we first determined the dynamics of viremia in the unvaccinated challenged control group ZIKV challenge. As shown in Fig.4A, the peak of viremia occurred at day 3 post-challenge, and started to drop at day 7. At day 10, viremia dropped to background level in the unvaccinated challenged control group. This is also consistent with previous observations that the ZIKV infection only caused transit viremia in immunocompetent mice. We have now completed the detection of the viremia at day 7 post-challenge (Fig.4C). The viremia in vaccinated groups (rVSV-G1670A-E, rVSV-G1670A-prM-E, rVSV-G1670A-prM-E-NS1, and pCI-prM-E) has been cleared, whereas ZIKV RNAs were still detectable in the unvaccinated control and rVSV-G1670A group. Therefore, it is not necessary to detect viremia after day 7 post-challenge.

Reviewer #2:

Introduction:

1. Pg 3: Authors should use the term “Congenital Zika Syndrome” which has been preferable adopted for description of the effects of Zika infection in the fetus and infants.

Response: Yes, it has been corrected.

2. Pg 6: Authors cited another paper that also tested the VSV expressing prM-E from ZIKAV in mice. They should discuss this article with more detail in the discussion section presenting what was different from the present manuscript.

Response: We have provided a detailed discussion on this paper. Please see lines 528-536.

Results:

3. Pg 7: The presentation of the Zika envelope protein structure is very poor. Authors should explain that it is composed of an ectodomain, consisting of three domains (I, II and III), and stem and transmembrane regions. They could cite the reference of Sirohi et al, Science 2016, which investigated the structure of the virus. Such explanation could perhaps be inserted in the introduction section and in results or materials and methods authors could indicate which part was deleted in truncate E proteins. Did they remove all stem and transmembrane regions?

Response: We have now modified the description of Zika envelope structure and cited the paper as suggested by the reviewer. This has been included in the Materials and Methods section. Yes, we removed all stem and transmembrane regions to construct E truncation mutants. Please see lines 157-160, and Fig.S1A.

4. In fig. S2 one band corresponding to the NS1 protein should also not be present in rVSV-prM-E-NS1 infected cells?

Response: Theoretically, we should be able to detect NS1 protein in rVSV-prM-E-NS1-infected cells using S³⁵ metabolic labeling experiment (Fig.S4). Because VSV N protein and ZIKV NS1 protein have a similar molecular weight, we are not able to separate them in SDS-PAGE in this experiment. Therefore, we performed Western blot analysis using NS1-specific antibody. As shown in Fig.1C, Fig.2B and E, and Fig.S3C, NS1 protein can be detected by Western blot, confirming that NS1 was expressed and cleaved from prM-E-NS1 polypeptide.

5. Pg 11: In the last sentence authors affirmed that results demonstrated that inoculation with rVSV-prM-E-NS1 induced high antibody levels 1 or 2 weeks after inoculation, but this is not what we observe in fig 3. As we see in fig 3, antibody levels in this group were similar to those detected in mice immunized with truncated E protein without prM.

Response: The review is correct. Mice in rVSV-prM-E-NS1 group had a similar level of antibody with those mice immunized with truncated E protein without prM (Fig.1F). The sentence has been modified.

Pg 15: Authors affirmed that results demonstrated that co-expression of NS1 enhances Th2 and Th17 responses but they should also comment that in this group the IFN-g was also higher comparing to the other groups (fig 6B), suggesting a Th1 response. It seems that the presence of NS1 lead to a more balanced response including both Th1 and Th2 cells.

Response: The reviewer is correct. Co-expression of NS1 had a higher IFN- γ response than other groups. The presence of NS1 leads to a more balanced response including both Th1 and Th2 cells. We have added this point to the Results Section. Please see lines 350-353.

Pg15/16: Authors affirmed that the mtdVSV-based vaccines were safe, but, as we saw in fig S4, animals immunized with rVSV-G1670A-prM-E also lost body weight. Overall, they should be more cautious in asserting about the safety of these viruses based on the present study.

Response: We thank you the reviewer for this question. Please see our response to Reviewer #1 about the weight loss for each experiment. Our study showed that mRNA cap methylation can serve as an approach to attenuate VSV to utilize as a vaccine vector. In the current study, we used G1670A mutation in the VSV L protein, which is only defective in G-N-7 methylation. To achieve more attenuation, we can use mutations in MTase active site (such as D1762A, K1651A, and E1833Q), which were defective in both G-N-7 and 2'-O methylation. In our previous publication, we showed that these recombinant viruses were completely attenuated in BALB/c mice (Ma et al., Journal of Virology, 2014). We have added this to the Discussion section (Please see lines 635-644).

Pg 17: Why did authors immunized A129 animals by the intramuscular route while BALB/c mice were inoculated intranasally? Is it because A129 are more susceptible and the intranasal route would lead to high morbidity and death?

Response: We thank the reviewer for this comment. Yes, A129 mice are much more susceptible to VSV infection than BALB/c mice. A dose of 50 PFU of VSV is lethal to A129 mice. After intranasal inoculation, VSV infects olfactory neurons in the nasal mucosa and subsequently enters the central nervous system (CNS) through the olfactory nerves and disseminates to the brain. Thus, when designing the experiment in A129 mice, we chose intramuscular route for vaccination to reduce the side effect. In addition, it was known that intramuscular route was effective for delivery of VSV into mice. Importantly, our results showed that 10^5 PFU (2,000 higher than the lethal dose) of rVSV-G1670A-prM-E-NS1 was completely attenuated in A129 mice, whereas rVSV-G1670A-prM-E still caused some weight losses. This result demonstrated that rVSV-G1670A-prM-E-NS1 was highly attenuated in A129 mice.

Standard deviations should be inserted in several graphs (Figs 3, 5A, S1, S3, S4, S5), informing it in figure legends.

Response: We have now added the standard deviations to all figures.

Discussion:

Pg 21: Authors compared the two immunization routes (intramuscular and intranasal) and suggested that it may play a role in the antibody response, by delaying it, for instance. However, in this case it was not only the inoculation route that could influence results but also the background of each animal (BALB/c and A129).

Response: The reviewer is correct. It is possible that the backgrounds of mice may affect the immune response. We have modified the sentence.

Pg 22: Authors compared the immune response induced by the DNA and VSV-based vaccines. It should also be interesting to evaluate the long-term response induced by both strategies.

Response: Our result showed that VSV-based vaccine candidates had a higher immunogenicity compared to the DNA vaccine, based on the short-term immunization experiments. In the future, we will compare the long-term immune response between DNA and VSV-based vaccine.

Reviewer #3:

1. “.....However, there are some weaknesses. First of all, there are way too many figures – this should be restricted to a maximum of 6 or 7 – a lot of the results can be consolidated and some of the results can be removed, for example the NS1 partial protection statement, which is not convincing and based on 1 mouse.....”

Response: We have consolidated the results into 7 figures. All other results are reported as supplementary figures (a total of 9). The finding that NS1 alone can provide partial protection could be of interest to the ZIKV and flavivirus field. As suggested by the reviewer, we have now generated rVSV-G1670A expressing NS1 alone (rVSV-G1670A-NS1) and compared the protection efficacy between rVSV-G1670A-NS1 and pCI-NS1 in BALB/c mice. We found that both rVSV-G1670A-NS1 and pCI-NS1 provided substantial protection against viremia in BALB/c mice. These results are shown in Fig.6E-H, lines 478-499. In combination of our previous data in A129 mice, it should be reasonable to conclude that NS1 alone can provide partial protection against ZIKV infection. In fact, our data is consistent with previous observations on NS1 of several of other flaviviruses.

Major comments and concerns:

Line 49-51: Based on current published data, neutralizing antibodies to the Zika envelope is essential for viral clearance. More importantly, a Zika vaccine without the envelope protein but containing the NS1 antigen only, has not been shown to protect against Zika challenge in this manuscript. This sentence needs to be removed from the abstract.

Response: I agree with the reviewer’s statement that neutralizing antibodies to the Zika envelope is essential for viral clearance. However, in addition to our experiment in A129 mice, our new data found that both rVSV-G1670A-NS1 and pCI-NS1 provided substantial protection against viremia in BALB/c mice (Fig.6E-H). Our data also showed that co-expression of NS1 with prM-E modulated the T cell and antibody responses. Taken together, NS1 played an important role in modulating ZIKV immune response. We have modified Lines 49-51.

Line 218,282,336: An intranasal route of immunization is not ideal, as it cannot be translated to humans. Why did the authors not do an intra-muscular immunization?

Response: The review is correct. Intranasal route of immunization may not be translated into human. However, it has been shown that VSV-based vaccine can be delivered into animals via many routes such as intranasal, intramuscular, oral etc. In fact, we showed that intramuscular vaccination of VSV-based vaccine in A129 also triggered a high level of protective immunity. This suggests that intramuscular vaccination is effective in triggering immune response. Please also see our response to Reviewer#2 about the vaccination route.

Line 213-214: Zika virus control would be required in order to conclusively draw similarities between the VLP’s made by the rVSV-Zika constructs and Zika virus. The EM is off low quality and the conclusion not supported by the presented data. Remove Figure 2E and the VLP statements from the manuscript.

Response: Yes, we have now included an EM image of purified Zika virus virions as a control. We also provided ZIKV VLPs images with higher magnification. **Fig.2E** is now listed as a supplementary Figure (**Fig.S5**). It is critical to demonstrate that the expression of ZIKV prM-E antigen by VSV can lead to the assembly of ZIKV VLPs, as they contain optimal epitopes for triggering immune response. Our EM analysis showed that expression of prM-E or prM-E-NS1 led to a high yield of ZIKV VLPs. We have now labeled the VSV and ZIKV VLPs in the images. We have also modified the VLP statements in the manuscript, as suggested by the reviewer. Please see lines 221-224.

Line 227-233: The authors have conducted pathogenicity experiments (intranasal inoculations) with their recombinant VSV-Zika vaccines and used weight loss as a measure of pathogenicity. Looking at figure S3, most mice except for the pCI-prME and DMEM controls show varying degree of weight loss, indicating that the constructs do show some pathogenicity at earlier time points. In order to conclusively prove that the recombinant vaccines are truly non-pathogenic, mouse brains should have been harvested at endpoints and analyzed by for viral VSV RNA.

Response: We have now analyzed the infectious VSV in brain tissue by plaque assay and VSV RNA by real-time RT-PCR in **Animal Experiments 3 and 4**. These results were reported in **Fig.4D and 5H**. Please see lines 390-397, and lines 442-447. Infectious VSV was not detected in all the brain samples in both experiments. VSV RNA was not detectable in BALB/c (**Fig.4D**) and A129 mice (**Fig.5H**) vaccinated with rVSV-G1670A-prM-E-NS1. However, VSV RNA can be detected in other VSV-ZIKV constructs. Our results showed that rVSV-G1670A-prM-E-NS1 is the most attenuated recombinant virus. For mouse body weight changes, please also see our response to Reviewer #1.

Figure 4 (A,B,C): Important controls - rVSV-prME and rVSV-prME-NS1 and rVSV are not in all the figures. Ideally these attenuated rVSV's should express less of the Zika E in comparison to the rVSV-prME.

Response: We have now compared the ZIKV E protein expression in parental VSV and VSV-G1670A vectors (**Fig.2D**). Please see lines 288-292. Recombinant rVSV-G1670A-prM-E-NS1 and rVSV-G1670A-prM-E had less ZIKV E protein expression compared to rVSV-prM-E-NS1 and rVSV-prM-E, suggesting that they are more attenuated.

Line 337: The dose and route of Zika challenge should be mentioned in the result, legend and the method section. The reader has to go through result, figure legend and method to get the complete information!

Response: Yes, we have now included the dose and route of Zika challenge in the Result, Figure Legend, and Materials and Methods section.

Line 361: It is important to show the complete clearance of Zika virus on day 7 and future time points for the vaccinated group and unvaccinated group. Why is this data not shown? The Zika

viral RNA data for different groups and different days could be consolidated into one graph replacing figure 7A and B.

Response: We agree with the reviewer. We have now provided the viremia data on day 7 post-challenge. Please see Fig.4C and lines 384-388. As shown in the Fig.4C, Zika virus has been completely cleared in the vaccinated groups, whereas the unvaccinated challenged controls still have detectable viremia. Please also see our response to Reviewer #1.

Figure 9A+9B can be consolidated into 1 graph i.e. week 1 and 3, Zika E IgG Response. Also mention in the legend that the titers shown are total IgG titers. Figure 9C+9D can be consolidated into 1 graph. Figure 9E and F can also be consolidated into 1 graph.

Response: We thank the reviewer for this suggestion. Since each panel contains a number of groups and needs to be statistically labeled, the figure becomes very complicated if we combine these panels. For clarity, we chose to keep them in separate panels in a single figure (Fig.5). We also indicated in the legend that the titers shown are the total IgG titers.

Line 387: The strain and route of Zika virus challenge should be mentioned. The reader has to go through the result, figure legend and method to get the complete information. Confusing!

Response: We have now clearly indicated the strain and route of Zika virus used for challenge.

Line 411-435: The experiment was terminated at day 7 and at this point the mice that were immunized with pCI-NS1 had viremia (in the brain, lung spleen and uterus), symptoms and encephalitis. Hence, it is difficult to conclude partial protection based on slightly reduced symptoms (symptoms are bias criteria in the first place). rVSV-G1670A-NS1 construct should have been used to study this. Why did the authors suddenly switch to the DNA vaccine platform to study this?. These data need to be removed, they are off topic.

Response: As suggested by the reviewer, we have now constructed rVSV-G1670A-NS1 and compared the protection efficacy with pCI-NS1 in BALB/c mice. These new results were reported in Fig.6E-H. Our main findings are (1) rVSV-G1670A-NS1 triggered significantly higher NS1-specific antibody than pCI-NS1; (2) both rVSV-G1670A-NS1 and pCI-NS1 provided substantial protection against viremia compared to pCI control group; (3) rVSV-G1670A-NS1 had a higher protection efficacy than pCI-NS1. At day 7 post-ZIKV challenge, viremia was cleared in mice vaccinated with rVSV-G1670A-NS1, whereas significant viremia was still detectable in pCI-NS1 group. Please see lines 478-499.

We also attempted to test rVSV-G1670A-NS1 in A129 mice. Unfortunately, rVSV-G1670A-NS1 still caused significant body weight loss in A129 mice. Therefore, we terminated this study. In the Discussion section, we have discussed the strategy to further attenuate rVSV-G1670A-NS1 in order to test it in A129 mice. Please see lines 635-644.

In the original submission, our results indicated that pCI-NS1 is capable of providing partial protection against ZIKV challenge in A129 mice. This is based on the fact that (1) pCI-NS1 showed less body weight loss, ZIKV associated symptoms, and encephalitis compared to the pCI control (**Fig.6A, B, and C, and Fig.7G**); and (2) pCI-NS1 group had statistically less viremia (in the brain, lung spleen and uterus) compared to the pCI control (**Fig.7B-F**). It should be noted that we used a very high dose (10^5 PFU) of ZIKV for challenge experiments in A129 mice. In the published literature, several ZIKV researchers have used 100 or 1000 PFU of ZIKV for challenge experiment in A129 mice (Lazear et al., 2016; Aliota et al., 2016). Since A129 is highly susceptible to ZIKV infection, the level of immune response induced by pCI-NS1 may not be strong enough to confer complete protection against a high dose of ZIKV challenge. Our future experiments will compare the ability of NS1 alone (delivered by VSV vector or DNA vaccine) to protect against different doses of ZIKV challenge in A129 mice.

Finally, the partial or even complete protection induced by NS1 alone has been reported for several of other flaviviruses such as West Nile virus, Dengue virus, and yellow fever virus.

Taken together, our data from both BALB/c and A129 mice support the conclusion that NS1 protein of ZIKV plays an important role in protecting against ZIKV infection. This finding should be of interest to the field of ZIKV and flavivirus. Therefore, we decided to keep these data in the current manuscript.

Additionally, the authors could have included VSV viral load in the brains of the AG129 mice immunized with Mtase defective- rVSV-Zika vaccines and challenged with Zika virus as a supplemental figure to rule out pathogenicity of the viral vectors in these immunocompromised mice.

Response: We have now included the data on VSV viral load in the brains of AG129 mice (**Fig.5H**). Please see lines 442-447. First, no infectious VSV was detected in mice vaccinated with rVSV-G1670A-prM-E-NS1 or in rVSV-G1670A-prM-E. Second, VSV RNA was undetectable in rVSV-G1670A-prM-E-NS1 group by real-time RT-PCR, whereas approximately 6 logs of VSV RNA were detected in rVSV-G1670A-prM-E. This data is consistent with the data on body weight change. Recombinant rVSV-G1670A-prM-E-NS1 did not cause any body weight losses in A129 mice, whereas rVSV-G1670A-prM-E still caused weight losses in early time points. These data demonstrated that rVSV-G1670A-prM-E-NS1 was significantly more attenuated than rVSV-G1670A-prM-E.

Line 445: NS1 providing partial protection cannot be concluded as the T cell responses were seen for the rVSV-G1670A-prME-NS1 construct and not for the pCI-NS1 DNA vaccine.

Response: We have modified this sentence. As described above, we have now provided new data on rVSV-G1670A-NS1. We showed that both rVSV-G1670A-NS1 and pCI-NS1 provided substantial protection against viremia in BALB/c mice (**Fig.6E-H**).

Line 721: Route of ZIKV challenge is not mentioned in the method section.

Response: Yes, we have now mentioned the route of ZIKV challenge in the Materials and Methods section

Minor comments:

Line 200: Re-label to figure S2. S2A doesn't exist.

Response: Yes, it is corrected.

Figure 3,5,9: legend and figure should mention that the ELISA titers shown are mean endpoint titers of 5 mice.

Response: We have mentioned this in the figure and legend.

Figure S2: requires a size marker

Response: A size marker has been indicated.

Figure S4: The weight's reported in the figure are mean body weights per group and should be mentioned in the figure legend.

Response: Yes, we have stated this in the figure legend.

Reviewers' comments:

Reviewer #1 (Remarks to the Author):

The authors have performed some additional studies, particularly with a newly generated rVSV-G1670A-NS1 construct, and addressed many of the reviewer's concerns. However, this has raised significant issues with some of the constructs used in the study. The authors include a leader sequence prior to the start of prM in all rVSV constructs that include the prM protein. But the E-only and NS1-only constructs should also be preceded by leader sequences, yet are not. The lack of a leader sequence likely explains why the full length E only construct is not detected even in the cell lysates of infected cells (Fig. 1D), as it is not being directed to the correct cellular compartment. The use of rVSV-E as a "negative control" is an odd choice, because the full length, membrane bound E protein would not be predicted to be a useful immunogen regardless (even if it had the proper leader sequence), because it would not be secreted. The rVSV-NS1 construct has the same issue; NS1 is not a cytoplasmic protein, but is directed into the ER lumen during polyprotein translation. With the exception of the prM-containing constructs, any protective effects of the NS1-alone constructs (or non-protection with the E-alone construct) used in the current study are based on expression of viral proteins in a manner that does not occur during virus replication. There is no description of the ZIKV sequences used to generate the pCI plasmid constructs, so it is assumed that the pCI-NS1 also lacks a leader sequence.

Additionally, there are too many statements that are just not supported by the data:

Lines 240-241. "Overall, rVSVs co-expressing prM and E/E truncation mutants were more attenuated in mice than rVSV expressing E/E truncation alone." This statement is not supported by the data in Figure S6.

Lines 381-383: "In contrast, mice that had been vaccinated with rVSV-G1670A-prM-E, rVSV-G1670A-prM-E-NS1, and pCI-prM-E were protected (Fig. 4B)." This is only somewhat true. Yes there is a level of protection, but all of the vaccinated groups had detectable viremia in at least some of the animals.

Lines 435-438: "Similarly, high level ZIKV was detected in the brain, uterus, lung, and spleen of the pCI control group whereas no ZIKV RNA was detected in any of these organs in the rVSV-G1670A-prM-E and rVSV-G1670A-prM-E-NS1 groups (Fig. 7C-F)." This is not true- there is detectable viremia above the limit of detection in all of these graphs.

Lines 577-579: "At day 7 post-challenge, no viremia was detected in rVSV-G1670A-NS1 whereas significant viremia was present in pCI and pCI-NS1." This is not true- there is detectable viremia for rVSV-G1670A-NS1 in Figure 6H, and the use of the word "significant" is questionable, as the pCI-NS1 group had viremia close to the limit of detection.

Lines 628-630: "In fact, rVSV-G1670A-prM-E-NS1 had a greater protective efficacy against viremia than rVSV-G1670A-prM-E at day 3 post-challenge in A129 mice (Fig. 7A)." But this ignores the fact that the difference was rather small, and was no longer observed at day 7,

when both groups now had detectable viremia.

Additional comments:

1) Paragraph starting on line 73: The description of the virus and subviral particle structure, as well as the association between prM and E is lacking. The prM protein is an integral part of both virions and subviral particles, and undergoes a cleavage event during virus maturation. The text reads as if prM exists only to allow proper folding of E. Additionally, subviral particles are formed *in vivo*, yet the text suggests that they are only recombinantly expressed (line 84). The expression of prM followed by truncated E results in the secretion of soluble E. The expected immunogens (subviral particles versus soluble E) are not acknowledged clearly throughout the manuscript. This is particularly evident in the lack of any Western Blot analysis for prM or M, which should be visible in the cell lysates and supernatants of the prM-E constructs.

2) The figures still lack clarity as to what the data represents. The methods now state that all experiments were done three times. This is too broad of a statement to apply to the various types of experiments presented within. Did individual experiments have replicates? Multiple reviewers commented on the lack of description regarding error bars. The authors replied that they amended the manuscript, but none of the figure legends state what the error bars denote (standard deviation? Standard error? 90% confidence intervals?). Many figures do not state whether the statistics indicate mean or median (Figure 2H and I, for example).

3) EM pictures now include ZIKV virions, but there is no description anywhere about this sample. What virus strain? What cells was it produced in?

4) Figure 3- Were mouse samples pooled or treated individually for T cell experiments? Not knowing this makes it impossible to determine what data is expressed on the graphs. Is it group averages calculated from individual mice? If they were repeated three times as the methods state, how is this reflected in the graph?

5) The limit of detection for the qRT-PCR graphs is not consistent (compare Figure 7A to Figure 6G, for experiments with ZIKV, and Figure 5H versus 4D for VSV qRT-PCR- all have different limit of detection lines). The limit of detection also seems very high (>4 logs for ZIKV!), and results are often graphed below the limit of detection as exact values. If it is below the detection level, a constant, such as ½ the limit of detection, should be used.

6) Statistics not clear:

Lines 279- 281- "Single-step replication curves showed that VSV-G1670A-E and rVSV-G1670A-prM-E had replication kinetics similar to rVSV-G1670A, whereas rVSV-G1670A-prM-E-NS1 had a significant delay (Fig. 2A)." What statistics were used to compare? What is the p-value?

Lines 466-467: "The other 4 mice in the pCI-NS1 group exhibited clinical signs but less severe than the pCI group (Fig. 6A)." Statistics? One of the mice has the same clinical

score.

7) The absolute value of P-values <0.05 should be provided.

Reviewer #2 (Remarks to the Author):

Authors have reviewed most of questions pointed by reviewers. However, the question concerning safety of this vaccines and body weight loss of animals inoculated with the recombinant VSV was not fully elucidated. The differences observed between figs 2F and S9 was not satisfactorily clarified and discussed.

Besides, authors included one reference concerning the use of NS1 as a protective antigen against dengue. However there are other references that have previously demonstrated the protective role of NS1 of other flavivirus (Schlesinger et al., J. Gen. Virol 1987), including DNA vaccines against dengue (Cost et al., Virology 2007), which would be more appropriated.

Reviewer #3 (Remarks to the Author):

A novel Zika virus vaccine expressing pre-membrane, envelope and NS1 proteins

The reviewer highly appreciates the authors efforts to satisfy most of the comments. However, the following comments should be considered.

Major Concern:

Figure 1E: As stated by the authors the ZIKV VLPs expressed by their VSV viruses are smaller than the native ZIKV virion. It is unclear how the authors claim that they are structurally similar to the ZIKV virion, considering the ZIKV virions are granular and not exactly spherical while the ZIKV VLPs made by the VSV is an opaque spherical structure. The authors should remove this sentence. A size bar is only on one panel. The figure is of poor quality and doesn't add a lot.

Figure 7: It is unclear why the authors have not provided Zika viremia in the immunized groups till necropsy. Their reasoning, claiming no viremia in the unvaccinated group at Day 7 is not satisfactory for terminating the experiment, as the immunized group could have developed delayed viremia. In some settings viral loads are increasing over time. The experimental time is just too short and animals should be observed for at least 21 days after challenge.

Responses to editor and reviewers' comments

Introduction: We thank the three reviewers for their careful review of our manuscript. All of them had constructive and thoughtful comments which were extremely helpful in improving the manuscript. It took us approximately three months to complete these new experiments, as reviewer #1 suggested that we make new recombinant VSVs with anchor C (leader sequence or signal peptide) for some constructs and test them in mice, and reviewer #3 requested that we monitor viremia for prolonged time (21 days after ZIKV challenge). We have now completed these experiments and modified the manuscript, addressing each point, as described below.

Reviewer #1:

Major comments:

1. "...But the E-only and NS1-only constructs should also be preceded by leader sequences, yet are not. The lack of a leader sequence likely explains why the full length E only construct is not detected even in the cell lysates of infected cells (Fig.1D), as it is not being directed to the correct cellular compartment."

Response: The reviewer raises a very good point. We have now constructed recombinant VSV expressing E, E404, E414, and E415 with a leader sequence (anchor C, signal peptide). These recombinant viruses are named rVSV-aE, a404, a414, and a415 to distinguish them from the original viruses. The construct diagrams and plaque size are shown in **Fig.S1**. We also compared the expression of E proteins by these recombinant viruses with and without the anchor C sequence. With the anchor C sequence, the expression of E/E truncation proteins in cell lysates was significantly increased compared to those without anchor C sequence (**Fig.1D**). In addition, the E truncations were secreted into cell culture medium whereas full-length E protein was not (**Fig.S3C**). **These new results are reported in Fig.1D, Fig.S1, and Fig.S3C. Please see our response to the comment 3 for the NS1 constructs.**

2. "The use of rVSV-E as a "negative control" is an odd choice, because the full length, membrane bound E protein would not be predicted to be a useful immunogen regardless (even if it had the proper leader sequence), because it would not be secreted. "

Response: To address this concern, we constructed another recombinant virus, rVSV-G1670A expressing E with an anchor C sequence. This recombinant virus was named rVSV-G1670A-aE (**Fig.S7**).

Subsequently, we performed a NEW animal study to test the immunogenicity of rVSV-G1670A-aE in mice compared to rVSV-G1670A-prM-E and rVSV-G1670A-prM-E-NS1. The results were reported in Fig.4C, D, E, F, and Fig.S12 and 13. Please see lines 415-430. Only 1 out of 5 mice in rVSV-G1670A-aE group had an E-specific antibody response at weeks 1-4, but all 5 mice had an E-specific antibody response at week 5 in this group. After challenge with ZIKV, rVSV-G1670A-aE had relatively lower viremia than the two control groups (rVSV-G1670A and Saline) but they were not statistically different ($P>0.05$). Similar to our previous experiments, rVSV-G1670A-prM-E and rVSV-G1670A-prM-E-NS1 provided protection against viremia. Collectively, these data demonstrated that immunogenicity of

rVSV-G1670A-aE is much lower than rVSV-G1670A-prM-E and rVSV-G1670A-prM-E-NS1, despite the fact that the anchor C sequence has been included in rVSV-G1670A-aE construct. These results are also consistent with published literature, showing that prM-E is much more immunogenic than E alone.

3. “ The rVSV-NS1 construct has the same issue; NS1 is not a cytoplasmic protein, but is directed into the ER lumen during polyprotein translation. With the exception of the prM-containing constructs, any protective effects of the NS1-alone constructs (or non-protection with the E-alone construct) used in the current study are based on expression of viral proteins in a manner that does not occur during virus replication. There is no description of the ZIKV sequences used to generate the pCI plasmid constructs, so it is assumed that the pCI-NS1 also lacks a leader sequence.”

Response: In the previous version of the manuscript, when we constructed the rVSV-G1670A-NS1 and pCI-NS1, anchor C sequence was included in these constructs, but we had not stated that in the figure legend. We have now corrected that oversight. To prove this, we attach the upstream primer (primer name: VSV-ZIKV-Anchor C-NS1 F) which is in 82 bp length including anchor C sequence and was used for PCR the NS1 to construct the rVSV-G1670A-NS1 and pCI-NS1 (please see the **Support Material**). This primer was ordered from Sigma on 06/29/2017 (up left side).

During revision of our manuscript, Brault et al., reported that intramuscular immunization of immunocompetent mice with the Modified Vaccinia Ankara (MVA) vector expressing NS1 (MVA-ZIKV-NS1) vaccine candidate provided 100% protection against a lethal intracerebral dose of ZIKV (strain MR766). However, their study did not evaluate the efficacy of MVA-ZIKV-NS1 in A129 mice which is more highly sensitive to ZIKV infection. In our study, we showed that rVSV-NS1 only provided partial protection in A129 mice. Brault’s study should not diminish the significance of our manuscript. The main emphasis of our manuscript is to include NS1 with prM-E for the ZIKV vaccine development, the resultant recombinant virus rVSV-G1670A-prM-E-NS1 was highly attenuated in BALB/c and A129 mice and provided complete protection against ZIKA challenge in these two animal models. The concept of combination of prM-E-NS1 is novel, because it combines the protective effects of E and NS1 proteins.

Additionally, there are too many statements that are just not supported by the data:

1. Lines 240-241. “Overall, rVSVs co-expressing prM and E/E truncation mutants were more attenuated in mice than rVSV expressing E/E truncation alone.” This statement is not supported by the data in Figure S6.

Response: We should have explained the weight loss experiment better. Mice inoculated with wild type rVSV were all killed by day 10. All mice inoculated with recombinant VSVs with ZIKV gene insertions survived, however, they experienced body weight losses, which usually occurred at day 7 post-inoculation. Fig.S6 has 12 groups and it is difficult to separate each group. The following Table summarized the body weight at day 7 post-inoculation. As you can see, rVSV with prM had less body weight loss compared to rVSV without prM, suggesting that rVSV with prM was more attenuated. Statistical analysis showed that rVSV-prM-E has significantly less body weight loss compared to rVSV-E

($P < 0.05$), and rVSV-prM-E414 had significantly less body weight loss compared to rVSV-E414 ($P < 0.05$). In general, this is consistent with VSV gene expression strategy: a large gene insertion results in more attenuation.

Table 1 Body weight loss at day 7 post-inoculation

Group	Body weight (%) at day 7	Statistical analysis (compare with and without prM), P value
DMEM	101.8	
rVSV	72.4	
rVSV-E404	77.8	
rVSV-prM-E404	82.6	Compared to E404, $P = 0.44230862$
rVSV-E414	78.6	
rVSV-prM-E414	86.3	Compared to E414, $P = 0.045038882$
rVSV-E415	75.6	
rVSV-prM-E415	79.1	Compared to E415, $P = 0.581109223$
rVSV-E	74.4	
rVSV-prM-E	89.1	Compared to E, $P = 0.020665417$
pCI-prM-E	102.8	
rVSV-prM-E-NS1	96.5	

2. Lines 381-383: “In contrast, mice that had been vaccinated with rVSV-G1670A-prM-E, rVSV-G1670A-prM-E-NS1, and pCI-prM-E were protected (Fig. 4B).” This is only somewhat true. Yes there is a level of protection, but all of the vaccinated groups had detectable viremia in at least some of the animals.

Response: We modified the sentence to “In contrast, viremia in most mice vaccinated with rVSV-G1670A-prM-E, rVSV-G1670A-prM-E-NS1, and pCI-prM-E groups were under the detection limit at day 3 (3 and 4 mice in rVSV-G1670A-prM-E and rVSV-G1670A-prM-E-NS1 had near detection limit level of viremia, respectively, and 1 mice in pCI-prM-E group had a high level of viremia) (Fig.4A and Fig.S11). In addition, viremia was under detection limit from days 7 to 24 in these groups (Fig.4A). These results suggest that mice vaccinated with these vaccine candidates were protected”. Please see lines 401-407.

It would be fair to state that perfect scenario in animal studies is rare. For rVSV-G1670A-prM-E, rVSV-G1670A-prM-E-NS1, and pCI-prM-E vaccinated groups, some animals had undetectable viremia (below the detection limit) whereas some had a low level of viremia (near the detection limit). This level of variability is normal, as real-time RT-PCR is a highly sensitive method particularly as the Ct value reaches the detection limit. Given the fact that this was a single dose of vaccine, the efficacy of rVSV-G1670A-prM-E and rVSV-G1670A-prM-E-NS1 would be considered high.

Reviewer #3 suggests that we should monitor the viremia for a prolong time. In this experiment (Animal experiment 3), we actually monitored the viremia until day 24 after ZIKV challenge (Except pCI-prM-E group which only monitored at days 3 and 7). We now showed the viremia level until 24 days (Fig.4A). At the same time, we showed the viremia data at day 3 as dot format (Fig.S11A), which indicates how many animals were above detection limit. In the previous version of manuscript, viremia data in BALB/c mice were shown as log₁₀ RNA copies/ml, whereas viremia data in A129 mice were shown as log₁₀ PFU

equivalent RNA/ml. To be consistent, we have now used log 10 PFU equivalent RNA/ml for all viremia data. Please also see our below response regarding the detection limit in blood and different tissues.

It should be noted that we repeated the animal experiment (see Animal experiment 4) during the revision. **The results were reported in Fig.4C, D, E, F, and Fig.S12 and 13. Please see lines 415-430.** These results were essentially similar to **Fig.4A and Fig.S11.**

3. Lines 435-438: “Similarly, high level ZIKV was detected in the brain, uterus, lung, and spleen of the pCI control group whereas no ZIKV RNA was detected in any of these organs in the rVSV-G1670A-prM-E and rVSV-G1670A-prM-E-NS1 groups (Fig. 7C-F).” This is not true- there is detectable viremia above the limit of detection in all of these graphs.

Response: We have modified this sentence. “Similarly, high level ZIKV was detected in the brain, uterus, lung, and spleen of the pCI control group whereas under or near detection limit level of ZIKV RNA was found in these organs in the rVSV-G1670A-prM-E and rVSV-G1670A-prM-E-NS1 groups”. Such variability is normal, when the Ct value is near detection limit. It is also consistent with several other ZIKV vaccine studies (for example, the mRNA vaccine study published by Richner et al., 2017, Cell 168, 1114–1125, Fig.3 panel G, H, I, and J). Please also see our response to the comment #2. **Again, perfect animal studies are rare.**

4. Lines 577-579: “At day 7 post-challenge, no viremia was detected in rVSV-G1670A-NS1 whereas significant viremia was present in pCI and pCI-NS1.” This is not true- there is detectable viremia for rVSV-G1670A-NS1 in Figure 6H, and the use of the word “significant” is questionable, as the pCI-NS1 group had viremia close to the limit of detection.

Response: We have modified this sentence to: “At day 7 post-challenge, mice in the rVSV-G1670A-NS1 and rVSV-G1670A-prM-E groups had no detectable viremia **(except one in rVSV-G1670A-NS1 group which was near the detection limit)** whereas mice in pCI and pCI-NS1 groups still had a significant level of viremia ($P < 0.001$) **(Fig.6H)**.” See lines 529-532.

5. Lines 628-630: “In fact, rVSV-G1670A-prM-E-NS1 had a greater protective efficacy against viremia than rVSV-G1670A-prM-E at day 3 post-challenge in A129 mice (Fig. 7A).” But this ignores the fact that the difference was rather small, and was no longer observed at day 7, when both groups now had detectable viremia.

Response: At day 3, the difference was small **but the viremia in rVSV-G1670A-prM-E-NS1 was statistically lower than rVSV-G1670A-prM-E ($P < 0.05$) (Fig.7A)**. It is true that both groups had a low level of viremia at day 7 **(Fig.7B)**. But, there were dramatic differences between vaccinated and pCI control group (see **Fig.7A and B**). **As requested by reviewer #3 and the editor, we now did a long-term study, monitoring mouse body weight and viremia for 21 days in A129 mice after ZIKV challenge. No body**

weight losses or viremia were observed for 21 day time period except that a near detection limit level of viremia at day 3 (Fig.8E). Please also see our responses to reviewer #3.

Additional comments:

1) Paragraph starting on line 73: The description of the virus and subviral particle structure, as well as the association between prM and E is lacking. The prM protein is an integral part of both virions and subviral particles, and undergoes a cleavage event during virus maturation. The text reads as if prM exists only to allow proper folding of E. Additionally, subviral particles are formed *in vivo*, yet the text suggests that they are only recombinantly expressed (line 84). The expression of prM followed by truncated E results in the secretion of soluble E. The expected immunogens (subviral particles versus soluble E) are not acknowledged clearly throughout the manuscript. This is particularly evident in the lack of any Western Blot analysis for prM or M, which should be visible in the cell lysates and supernatants of the prM-E constructs.

Response: We have modified this sentence and a description of prM was added. We have tried to perform the Western blot using antibody against prM. Unfortunately, the antibody did not work in our assay. However, our Western blot against E and/or NS1 antibody clearly demonstrates that prM-E, and prM-E-NS1 was cleaved (Fig.1 and 2, Fig.S3, 4, 8).

2) The figures still lack clarity as to what the data represents. The methods now state that all experiments were done three times. This is too broad of a statement to apply to the various types of experiments presented within. Did individual experiments have replicates? Multiple reviewers commented on the lack of description regarding error bars. The authors replied that they amended the manuscript, but none of the figure legends state what the error bars denote (standard deviation? Standard error? 90% confidence intervals?). Many figures do not state whether the statistics indicate mean or median (Figure 2H and I, for example).

Response: Thank you. We strictly followed the standard requirement for biological experiments. In the Figure Legend, we now clearly state the number of replicates for EACH experiments and how data are presented. In general, there are two types of experiments in this manuscript. **First, *in vitro* experiments** such as Western blot, virus growth curve which were done in three independent experiments. The Western blot gels presented are a representative of three independent experiments. Virus growth data were performed in three independent experiments and are expressed as geometric mean titers (GMT) \pm standard deviation, and we state this. **Second, *in vivo* animal experiments.** For animal experiments, 5 or 10 mice were used/group to attain the power for statistical analysis. Data of body weights and T cell responses were the mean of 5 mice and expressed as mean \pm standard deviation. Antibody titers and viral RNA titers were geometric mean titers (GMT) of 5 or 10 mice \pm standard deviation.

3) EM pictures now include ZIKV virions, but there is no description anywhere about this sample. What virus strain? What cells was it produced in?

Response: ZIKV Cambodian strain (FSS13025) was used in all experiments including the Zika genes inserted into the VSV-based ZIKV vaccine, EM analysis, Zika virus challenge experiments, and virus-neutralizing assays. ZIKV Cambodian strain was grown in Vero cells, and purified by standard ultracentrifugation procedure. Please see lines 729-732, and lines 835-840.

4) Figure 3- Were mouse samples pooled or treated individually for T cell experiments? Not knowing this makes it impossible to determine what data is expressed on the graphs. Is it group averages calculated from individual mice? If they were repeated three times as the methods state, how is this reflected in the graph?

Response: We should have explained this experiment better. For T cell assays, each mouse sample was treated individually. After euthanasia, the spleen was isolated from each mouse, homogenized, a cell suspension prepared, split into three wells (triplicate per mouse) in 96-well microtiter plates (pre-coated with 20 µg/ml of ZIKV E protein), and cultured for 5 days. Flow cytometry was performed after intracellular staining with the corresponding anti-cytokine. For each group, data were expressed as mean % positive cells (the mean of 15 samples: 3 wells × 5 mice) ± standard deviation. These have been clearly stated in figure legend for Fig.3.

5) The limit of detection for the qRT-PCR graphs is not consistent (compare Figure 7A to Figure 6G, for experiments with ZIKV, and Figure 5H versus 4D for VSV qRT-PCR- all have different limit of detection lines). The limit of detection also seems very high (>4 logs for ZIKV!), and results are often graphed below the limit of detection as exact values. If it is below the detection level, a constant, such as ½ the limit of detection, should be used.

Response: Yes, limit of detection is different in different tissues, because RT-qPCR can be interfered by many factors such as different host RNAs, inhibitors, and so on. Similar observations have been reported for several other ZIKV studies (for example, the ZIKV mRNA vaccine study published by Richner et al., 2017, Cell 168, 1114–1125, Fig.3 panel G, H, I, and J). For all RNA quantification experiment, we used the SYBR Green-based qPCR method (TaKaRa, Japan) which may have a relatively lower sensitivity than specific probe-based qPCR method. Also, the detection limit for each animal experiment may be slightly different because we determined the detection limit for each experiment and each animal experiment had its own normal control (unvaccinated unchallenged). In addition, we have used a constant value in all figures if a value is below the detection level.

In the previous version of manuscript, viremia data in BALB/c mice were shown as log₁₀ RNA copies/ml, whereas viremia data in A129 mice were shown as log₁₀ PFU equivalent RNA/ml. To be consistent, we have now used log₁₀ PFU equivalent RNA/ml for all viremia data (**Fig.4A**), which has been reported by many other publications. Fig. S11 showed both log₁₀ RNA copies/ml (Fig.S11A) and log₁₀ PFU

equivalent RNA/ml (Fig.S11B) for comparison. The methods for calculating 10 PFU equivalent RNA/ml were described in lines 1036-1044.

6) Statistics not clear:

Lines 279- 281- “Single-step replication curves showed that VSV-G1670A-E and rVSV-G1670A-prM-E had replication kinetics similar to rVSV-G1670A, whereas rVSV-G1670A-prM-E-NS1 had a significant delay (Fig. 2A).” What statistics were used to compare? What is the p-value?

Response: Recombinant rVSV-G1670A-prM-E-NS1 was more attenuated than VSV-G1670A-E and rVSV-G1670A-prM-E., forming smaller plaques (**Fig.S7B**), displaying delayed cytopathic effects (data not shown) and delayed viral release at 8, 12, and 18 h post-inoculation compared to rVSV-G1670A-prM-E. The statistical analysis was done using ANOVA. The P value at 8, 12, and 18 h was 5.82173×10^{-09} , 0.002081238, and 0.111494403 respectively. So, there was significant reduction in viral release at 8 and 12 h post-infection. See lines 297-298.

Lines 466-467: “The other 4 mice in the pCI-NS1 group exhibited clinical signs but less severe than the pCI group (Fig. 6A).” Statistics? One of the mice has the same clinical score.

Response: We have modified this sentence. “3 out of 4 mice in the pCI-NS1 group had clinical score of 3 and 1 mouse in this group had a score 4, whereas all mice pCI group had a score of 4. There was no statistical difference between these two groups ($P > 0.05$).” See lines 501-502.

7) The absolute value of P-values <0.05 should be provided.

Response: The exact P-value is now indicated in all figure legends.

Reviewer #2:

1. Authors have reviewed most of questions pointed by reviewers. However, the question concerning safety of this vaccines and body weight loss of animals inoculated with the recombinant VSV was not fully elucidated. The differences observed between figs 2F and S9 was not satisfactorily clarified and discussed.

Response: Thank you for pointing this out. We have carefully checked the statement regarding the safety of the VSV constructs. Yes, we observed some differences in body weight between Figs 2F (T cell assay) and Fig.S9 (ZIKV challenge). For example, rVSV-G1670A-prM-E caused only 2.7% of body weight loss in Fig.2F, whereas the same virus caused approximately 8% body weight loss in Fig.S9. **First of all**, these two Figures were from two independent animal experiments. In Fig.2F, we used female mice (5 per group). In Fig. S9, we used 5 male and 5 female mice (10 per group). **Second**, the mouse age used in

Fig.S9 was two weeks younger than **Fig.2F**. Six-week-old female mice were used in Fig.2F. Four-week-old female and male mice were used in **Fig.S9**. Younger mice are more susceptible to VSV infection.

During the revision, we conducted a NEW animal experiment (Animal Experiment 4), which repeated a similar experiment three times. In this experiment, 6-week-old female mice were used. The body weight was shown in **Fig.S12**, rVSV-G1670A-prM-E had about 4% weight loss at day 7, and quickly recovered. No body weight losses were observed for rVSV-G1670A-prM-E-NS1 in all three experiments (**Fig.2F, Fig.S9, and Fig.S12**).

It should be emphasized that the final and most significant construct in this manuscript is the rVSV-G1670A-prM-E-NS1. Inoculation of 10^6 or 10^5 pfu of this recombinant virus did not exhibit any body weight loss in either immunocompetent BALB/c mice (**Fig. 2F, Fig.S9, Fig.S12**) or immunodeficient A129 mice (**Fig.5A and Fig.8A**), demonstrating that rVSV-G1670A-prM-E-NS1 is highly attenuated and safe in mice, as we know that 50 pfu of wild type VSV is sufficient to kill A129 mice.

2. Besides, authors included one reference concerning the use of NS1 as a protective antigen against dengue. However there are other references that have previously demonstrated the protective role of NS1 of other flavivirus (Schlesinger et al., J. Gen. Virol 1987), including DNA vaccines against dengue (Cost et al., Virology 2007), which would be more appropriated.

Response: Thank you. We have now discussed and cited these two references.

Reviewer #3:

1. Figure 1E: As stated by the authors the ZIKV VLPs expressed by their VSV viruses are smaller than the native ZIKV virion. It is unclear how the authors claim that they are structurally similar to the ZIKV virion, considering the ZIKV virions are granular and not exactly spherical while the ZIKV VLPs made by the VSV is an opaque spherical structure. The authors should remove this sentence. A size bar is only on one panel. The figure is of poor quality and doesn't add a lot.

Response: The sentence has been removed. A size bar has been added to all panels. Since in rVSV-prM-E or rVSV-prM-E-NS1 produced two types of particles (VSV and ZIKV VLP), it is difficult to completely separate ZIKV VLP from VSV. The VSV genomic RNA is encapsidated by the viral N protein and forms a helical ribonucleoprotein (RNP) complex. During purification, it is difficult to eliminate the RNP complex (helical or ring-like structure). However, the EM pictures clearly showed that ZIKV VLP can be found in the supernatant. This is the purpose of the EM analysis.

2. Figure 7: It is unclear why the authors have not provided Zika viremia in the immunized groups till necropsy. Their reasoning, claiming no viremia in the unvaccinated group at Day 7 is not satisfactory for terminating the experiment, as the immunized group could have developed delayed viremia. In some

settings viral loads are increasing over time. The experimental time is just too short and animals should be observed for at least 21 days after challenge.

Response: We have performed a NEW animal immunization experiment in immunodeficient A129 mice. These results are reported in **Fig.8**. Briefly A129 mice were immunized with 10^5 PFU of rVSV-G1670A-prM-E-NS1 or saline, animals were challenged with ZIKV at week 5 post-immunization. **After challenge, body weight and viremia were monitored for 21 days (as suggested by the reviewer and editor)**. As shown in **Fig.8A**, no body weight losses were observed after vaccination with rVSV-G1670A-prM-E-NS1, demonstrating that rVSV-G1670A-prM-E-NS1 was highly attenuated in A129 mice. High titers of E-specific and NS1-specific antibody were detected in the rVSV-G1670A-prM-E-NS1 group (**Fig.8B and C**). After challenge with ZIKV, there was no significant difference in body weight change between rVSV-G1670A-prM-E-NS1 and the normal control group (unimmunized unchallenged group) during the 21 day time period ($P>0.05$) (**Fig.8D**). In contrast, all mice in saline group (unimmunized challenged group) were dead at day 7 post-challenge (**Fig.8D**). Viremia was monitored every 3 days until day 21. No viremia was detected in rVSV-G1670A-prM-E-NS1 immunized group for 21 day except that a near detection level of viremia was detected at day 3 after challenge (**Fig.8E**). These data, in combination with our previous data (**Figs.6 and 7**), **strongly suggest that rVSV-G1670A-prM-E-NS1 was safe and provided complete protection against ZIKV challenge**.

In the Animal Experiment 3 (original **Fig.4**), we initially determined the level of viremia at days 3 and 7 after challenge. In that experiment, we actually collected blood samples every 3 days until day 24 after challenge (except the pCI-prM-E group which was only collected at days 3 and 7). **We have now run real-time RT-PCR for blood samples from days 10-24**. Viremia was under detection limit after 7 day in these vaccinated groups. Therefore, we merged original Fig.4A, B, and C into one panel (**Fig.4A**), which describes the viremia up to day 24 for all groups (except pCI-prM-E which is on days 3 and 7). At the same time, we also showed the viremia data at day 3 as dot format (**Fig.S11A**), which indicates how many animals were above detection limit. Please see our response to reviewer #1 (additional comment #2).

REVIEWERS' COMMENTS:

Reviewer #1 (Remarks to the Author):

The authors have generated new vaccine constructs and performed additional animal experiments, greatly strengthening the paper. Importantly, they have ensured that all constructs expressing ZIKV E and E truncations encode a leader sequence when appropriate. This begs the question as to why data obtained with the original constructs lacking a leader sequence are still included, particularly the Western Blots. As the paper is somewhat lengthy, I would suggest removing these results. Much of the data presented for the constructs designed to express soluble E could also be removed, as these were never tested in the mouse challenge models and do not add much to the paper. While I believe the manuscript could be shortened and focused more on the constructs encoding NS1, overall the authors have appropriately responded to the reviewer's comments.

Reviewer #2 (Remarks to the Author):

Authors have worked hard to answer questions pointed by referees. They have performed several other experiments including the construction of other recombinant VSV encoding full length E gene, as well as truncated sequences of this gene, all fused to the signal peptide sequence corresponding the C-terminal region of the C protein. They showed that rVSV construct with the anchor C signal peptide led to more abundant expression of E/E truncations compared to constructs without this peptide sequence and, more important, E truncated proteins were secreted to extracellular medium. However, no experiments were performed in order to evaluate the immunogenicity of the rVSV E truncated constructions or their protective potential! I understand that the manuscript is already extensive and results indicated that the construction with prM-E-NS1 seemed to induce better response comparing to others, but the information about E truncations that are secreted to extracellular medium is missing. It's possible that secretion of E protein, without the C-terminal sequence that is highly hydrophobic and in the context without prM, induces more neutralizing antibodies and could be more protective.

Minor comments:

It would be good if authors include one table summarizing all constructions, either with rVSV or DNA vaccines, antibody responses and protection. This will organize information presented in the manuscript that is described separately in the different figures.

In pg 9, line 197, authors seemed to be surprised with secretion of NS1 by the rVSV-prM-E-NS1. However, was this not expected, since the natural signal peptide of NS1 that is presented in the C-terminal sequence of E protein was included in this construction? They should comment it.

In the legend of Fig. 6, authors could include what were the criteria used to consider the

different scores of clinical signs (1 to 4).

Response to reviewers' comments

Reviewer #1:

“The authors have generated new vaccine constructs and performed additional animal experiments, greatly strengthening the paper. Importantly, they have ensured that all constructs expressing ZIKV E and E truncations encode a leader sequence when appropriate. This begs the question as to why data obtained with the original constructs lacking a leader sequence are still included, particularly the Western Blots. As the paper is somewhat lengthy, I would suggest removing these results. Much of the data presented for the constructs designed to express soluble E could also be removed, as these were never tested in the mouse challenge models and do not add much to the paper. While I believe the manuscript could be shortened and focused more on the constructs encoding NS1, overall the authors have appropriately responded to the reviewer's comments.”

Response: We thank the reviewer for this suggestion. We constructed a total of 14 VSV constructs with wild-type VSV backbone, 5 VSV constructs with G1670A backbone, and 2 DNA vaccine constructs. All of them have been tested in either BALB/c and/or A129 mice except for rVSV-aE404, aE414, and aE415. Removal of Western blot data would be involved in cropping some gels. As suggested by editor, it is better to keep these data. In addition, most of data regarding characterization of these constructs have been attached as supplementary Figures. As also suggested by the editor, we have shortened the Introduction and Discussion sections.

Reviewer #2:

1. “.....They showed that rVSV construct with the anchor C signal peptide led to more abundant expression of E/E truncations compared to constructs without this peptide sequence and, more important, E truncated proteins were secreted to extracellular medium. However, no experiments were performed in order to evaluate the immunogenicity of the rVSV E truncated constructions or their protective potential! I understand that the manuscript is already extensive and results indicated that the construction with prM-E-NS1 seemed to induce better response comparing to others, but the information about E truncations that are secreted to extracellular medium is missing. It's possible that secretion of E protein, without the C-terminal sequence that is highly hydrophobic and in the context without prM, induces more neutralizing antibodies and could be more protective.”

Response: Thank you for this comment. In this manuscript, we have tested all VSV constructs and DNA vaccine constructs in animals except the rVSV E truncated constructions with a leader sequence (rVSV-aE404, aE414, and aE415). Although it would be interesting to determine whether they can induce more neutralizing antibodies in animals, the main emphasis of this manuscript is the construction of rVSV-G1670A-prM-E-NS1 and the role of NS1 in protection. We have clearly showed that rVSV-G1670A-prM-E-NS1 has excellent safety, immunogenicity, and protection in various animal models. In addition, we have shown that NS1 modulates immune response and provide partial protection. In the future, we will determine the immunogenicity of three rVSV E truncated constructions in animal models.

2. It would be good if authors include one table summarizing all constructions, either with rVSV or DNA vaccines, antibody responses and protection. This will organize information presented in the manuscript that is described separately in the different figures.

Responses: we have provided a supplementary Table 1, summarizing all constructs (including both VSV and DNA vaccines).

3. In pg 9, line 197, authors seemed to be surprised with secretion of NS1 by the rVSV-prM-E-NS1. However, was this not expected, since the natural signal peptide of NS1 that is presented in the C-terminal sequence of E protein was included in this construction? They should comment it.

Response: We have deleted “interestingly” from this sentence.

4. In the legend of Fig. 6, authors could include what were the criteria used to consider the different scores of clinical signs (1 to 4).

Response: We have now included the detailed criteria for scoring the clinical signs in legend of Fig.6.